# Evidence for Eocene aridification of the Atacama Desert's hyperarid core

**Benedikt Ritter-Prinz** [1] ✉, **Steven A. Binnie** [1], **Finlay M. Stuart**[2], **Derek Fabel** [2], **Richard Albert** [3], **Volker Wennrich** [1] **& Tibor J. Dunai** [1]

The Atacama Desert is the most arid non-polar region on Earth, yet the timing and drivers of its hyperaridity remain debated. The earliest record of extreme aridification is preserved in the Coastal Cordillera of Northern Chile at the Oligocene-Miocene boundary. However, clast exposure ages on low-relief surfaces and supergene mineralisation ages suggest that low precipitation, and thus limited surface activity and weathering, may have been established earlier. To test the Miocene hyperaridity hypothesis, we have established a record of surface activity based on cosmogenic $^{21}$Ne concentrations in 135 locally-derived quartz clasts from low-relief surfaces in the desert's core. Thirty-two clasts have modelled exposure durations of Oligocene age or older. Their long-term surface preservation suggests exceptionally low landscape evolution rates and implies that aridification initiated earlier than the development of the Humboldt Current and major Andean uplift. We hypothesize that global cooling following the Early Eocene Climatic Optimum was likely a key driver of regional aridification.

Transitions to extreme climatic states are linked to interaction between astronomical forcing, global climate, oceanic circulation and regional tectonics[1–8]. The Atacama Desert in Northern Chile is one of the oldest and driest deserts on Earth (Fig. 1), it is an extreme habitat for life on Earth serving as a commonly-used analogue for the Martian surface[9,10], and is characterised by exceptionally low rates of erosion and sedimentation[11–13]. Easterly-derived moisture is the primary source of precipitation at the western Andean foothills, where precipitation declines from more than 300 mm/yr at 5000 m altitude to less than 20 mm/yr at 2300 m[14] (Fig. 1C). Extreme hyperarid conditions prevail below 2300 m, with an mean annual precipitation in the region of less than 2 mm/yr[14], especially in the Chilean Coastal Cordillera (Fig. 1C). When and why the region attained hyperaridity is not well established[15–20]. The paucity of datable paleoclimate archives in the sedimentary record has hampered the development of a complete understanding of the Cenozoic evolution of the region.

The sediment record from the Atacama Desert points to arid conditions since 150 Ma[21]. The exceptionally high concentration of cosmogenic $^{21}$Ne in locally-derived quartz clasts from low-relief surfaces in the Coastal Cordillera of northern Chile[15,16,22] preserve the earliest record of extreme aridification (Early Miocene) that appears to coincide with the establishment of the Humboldt Current some-time during the late Oligocene-Early Miocene and the uplift of the Andes during the Miocene[15,17,22–24]. $^{40}$Ar/$^{39}$Ar ages of supergene mineralisation from the arid/hyperarid Andean Precordillera record the decrease of precipitation rates below the threshold (100-120 mm/year) necessary for deep leaching of metals from porphyry copper deposits around the same time[24–26]. Paleosols from the Precordillera and sedimentological archives from the Andean foothills, however, record a switch to hyperarid conditions during the Late Miocene[19,27] that coincides with the development of the high altitude Altiplano and implies that the development of an intensely arid climate was not synchronous across the region[16,24].

A small number of quartz clasts from widely dispersed very low relief surfaces in the Coastal Cordillera record Late Oligocene exposure ages determined from cosmogenic nuclide concentrations[15,16,22]. This implies that landscape stagnation occurred significantly earlier than the prevailing models predict, and implies that the transition to

[1]Institute of Geology & Mineralogy, University of Cologne, Cologne, Germany. [2]Scottish Universities Environmental Research Centre, East Kilbride, UK. [3]Frankfurt Isotope and Element Research Center (FIERCE), Goethe-Universität Frankfurt, Frankfurt, Germany. ✉e-mail: benedikt.ritter@uni-koeln.de

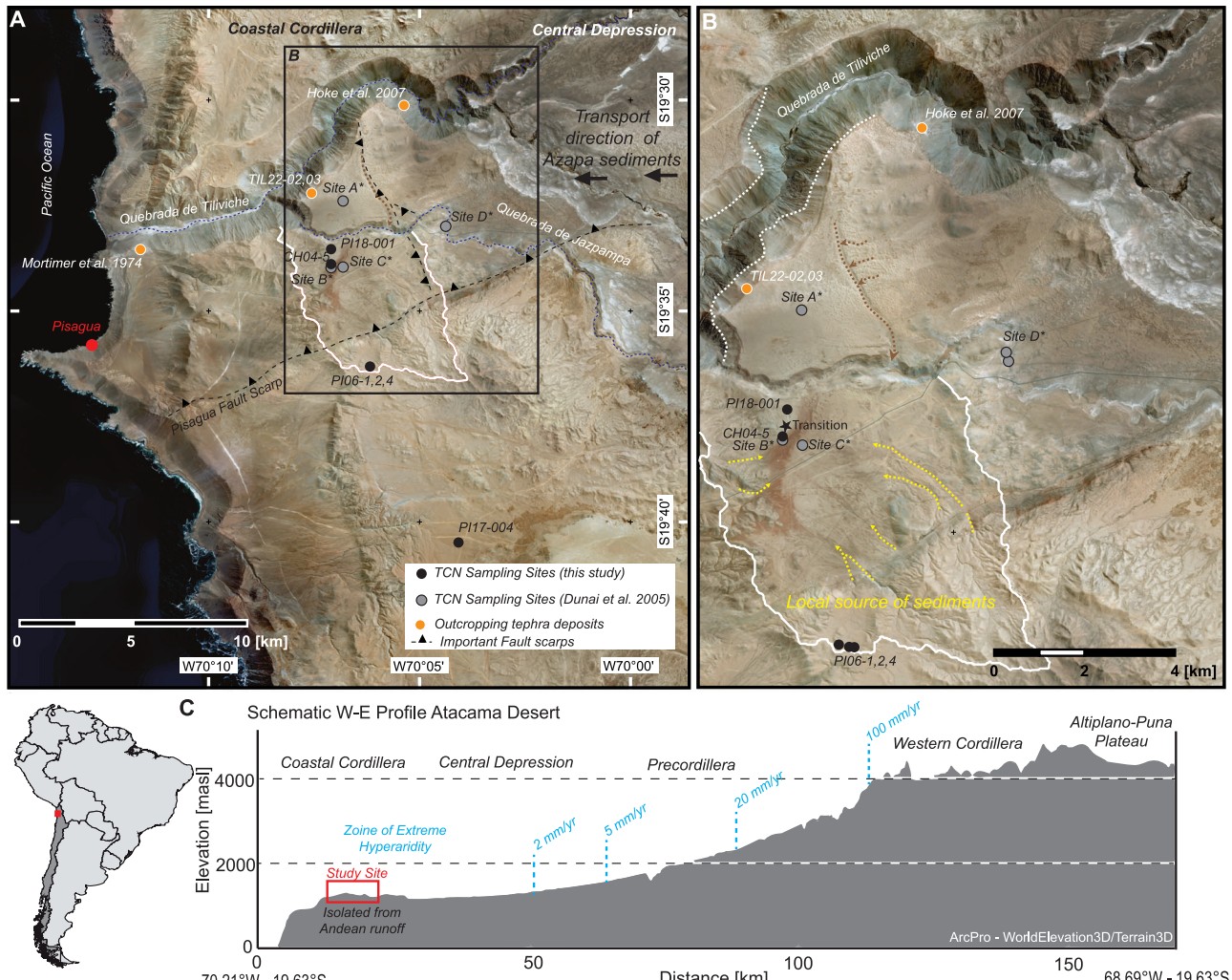

**Fig. 1 | Overview of the study area.** Satellite imagery (Earthstar Geographics SIO, ©2023 MAXAR) of the study area in northern Chile. (**A**) Overview of the study area located in the Coastal Cordillera of the northern Atacama Desert south of Arica. The white line marks the drainage catchment (~34 km²) of the sampled depositional surface. The sampled surface is bounded to the north by the deeply incised Quebrada de Tiliviche and protected from influence from the east by an uplifted fault scarp. The grey circles are sampling sites published by Dunai, et al.[15]. The black dots indicate the new sampling sites in this study. The Quebrada de Jazpampa incised into the surface and caused Site A to be isolated from the depositional system. PI17-004, to the south of the main study area, is located in the higher-elevated area of the Coastal Cordillera and may contain a clast population that has not been affected by erosion, such as samples PI06-1, 2 and 4. The orange dots mark outcropping tephra deposits studied by Mortimer, et al.[43] and Hoke, et al.[77] and sample sites TIL22-02, 03. (**B**) Close-up of the sampling locations. Water and entrained sediment move along the stippled arrows (based on satellite imagery and verified by field observations). Yellow stippled lines indicate the transport of vein quartz clasts from potential source areas. The white dashed lines indicate the basal contact (bedrock) of the Azapa Formation sediments[15]. The Quebrada Tiliviche and Quebrada de Jazpampa dissected these sediments following regional uplift. (**C**) Map of South America indicating the study site in red. Topographic west-east profile through the Atacama Desert created using ArcPro – WorldElevation3D/Terrain3D data. Indicated is the division of the Atacama Desert into its geographic units. Blue lines mark extrapolated recent mean annual precipitation modified from ref. 14.

hyperaridity may have been earlier. Occasionally pre–Miocene $^{40}Ar/^{39}Ar$ ages have been recorded for secondary minerals from supergene copper deposits from the Precordillera[24,25,28–31]. These hints that tectonic events, such as the uplift of high-altitude Andes and growth of the Altiplano, may have intensified regional aridification[24], but they may not have been responsible for the initiation of the intense aridification of the hyperarid core of the Atacama Desert.

The relict low-relief surfaces of the Coastal Cordillera in Northern Chile are ideally suited for studying the long-term climate history of the Atacama Desert[15–17,22,32]. Here we use cosmogenic $^{21}Ne$ concentrations to determine the exposure duration of 122 locally-derived quartz clasts from a suite of interconnected surfaces in northern Chile in order to determine whether the hyperarid climate of the Atacama Desert was established during the Miocene or earlier. We also revisit the study site originally investigated by Dunai, et al.[15], substantially

expanding their dataset by integrating previously published data with new $^{21}Ne$ and $^{10}Be$ measurements from multiple locations, as well as U-Pb zircon tephrochronology; in total, we compile concentrations from 135 $^{21}Ne$ and seven $^{10}Be$ surface clasts. We show that many clasts have been at the surface since the Mid Eocene. They cannot have originated from high altitudes in the uplifting Andes and require that water-limited conditions were established in the Chilean Coastal Cordillera significantly earlier than previously recognised.

The extreme hyperarid core of the Atacama Desert lies between 19–23°S[16], comprising the western parts of the Central Depression and the Coastal Cordillera. The Coastal Cordillera is a topographically isolated Jurassic magmatic arc bounded by the Pacific Ocean to the west and the Central Depression to the east. Elevation increases east of the Central Depression towards the Precordillera and high Western Cordillera (Fig. 1A, C). Uplift of the Coastal Cordillera initiated during

the Oligocene–Miocene transition and is thought to have isolated it from the Central Depression and thus the supply of Andean-sourced sediments in this part of the Andes[15,33–35].

Slow erosion and weathering, coupled with limited diffusive sediment transport (e.g. soil creep and hillslope transport by frictional and gravitational forces), generated by prolonged arid to hyperarid conditions for millions of years has reduced relief and filled depressions in the Coastal Cordillera and parts of the Central Depression[36]. Gypsum salt crusts and nodules formed by atmospheric deposition are common[27,37–39] and testify to the antiquity and low surface activity of the landscape.

## Results and Discussion
### Evidence of predominantly arid to hyperarid conditions
We have measured the concentration of cosmogenic $^{21}$Ne in angular gravel-sized vein quartz clasts (Supplementary Information Fig. S1) from several low-relief alluvial surfaces above the lower reaches of the Quebrada de Jazpampa in the Coastal Cordillera (Fig. 1, Supplementary Information Fig. S2). The surfaces are ~900 m above sea level (m a.s.l.) and have previously been shown to host quartz clasts with exceptionally long surface exposure histories[15]. The surfaces sit on thin alluvial deposits that overlie several volcanic ash (tephra) layers. The ashes cap the Azapa Formation (rounded fluvial gravels, Supplementary Information Figs. S3, S4), a regionally thick sediment sequence that was deposited around the Oligocene-Miocene transition[15,33–35](Fig. 1A, B). The thick Azapa Formation strata are linked to the synchronous deposits of the Lower Moquegue Formation, the lower part of the Altos de Pica Formation, the Tambores Formation, and the Calama Formation[17,40]. These deposits are the erosion products from the uplifting Andes to the east[41] and were predominantly deposited within the Central Depression, where they partially infilled basins and depressions that drained to the Pacific Ocean[15,42]. These sediments record a non-local (allochthonous) climate signal. K-Ar dating of a tephra near the Pacific coast that is considered to be equivalent to the above-mentioned Jazpampa tephra layers yielded an age of $21.8 \pm 0.3$ Ma[43] (recalculated using[44] and[45]). The Azapa and tephra deposits are overlain by low-relief surfaces that have been effectively protected from runoff from the Precordillera and the emerging Andes to the east since deposition (Fig. 1A, B). Consequently, the quartz clasts sampled from site CH04-5 (Fig. 1B) must have originated from catchments within the Coastal Cordillera[15], likely from veins in bedrock exposed on a local topographic high to the south (Fig. 1B, C, yellow dashed line). In this scenario, the transport of the clasts relied exclusively on local precipitation[15]. Episodic sediment transport in response to rare, localised torrential rain events is well documented in the Atacama region[46,47]. The sample area of CH04-5 is located immediately above a transition, where the $CaSO_4$-crust-covered valley floor changes into a series of salt-karst depressions that dominate the valley floor further downstream[15] (Fig. 1B). Most clasts were collected from the top of a friable $CaSO_4$ crust that covers the sediment surface. Several quartz clasts have been sampled from the bottom of a salt karst depression located approximately 450 m downstream of the transition (PI18-001; Fig. 1B).

Further quartz clasts were sampled from two low-relief surfaces to the south (sites PI-06 and PI17-04, Fig. 1A, Supplementary Information Fig. S5). Site PI-06 is the headwater of the catchment that has fed clasts to the Jazpampa surfaces (Fig. 1). Surface PI17-04 is a similar area from the same isolated topographic high. Both sites are locally the highest topographic points, typically ~300 m higher than the Jazpampa surfaces, and are rimmed by outcrops of Jurassic volcanic rocks[48] that are the source of the clasts. Both sample sites provide information on the exposure of clasts from the isolated source region within the Coastal Cordillera, which are not affected by sediment transport from the Andes to the east.

The clasts are what remains of sheet-flow deposits after the fine material has been removed by deflation and the lithic fragments destroyed by salt weathering, which is a consequence of the intense coastal fog (camanchaca)[49]. Rare rock clasts are characterised by a high degree of weathering and friability. The single-crystal angular vein quartz clasts (Supplementary Information Fig. S1) collected for this study are resistant to salt weathering and wind erosion. The source-distant location safeguards sediment mixing during transport by recurrent sheet-flows. Similar quartz clasts are found in sub-catchments to the south and southeast (Fig. 1B) demarking the local source of the clasts.

Cosmogenic $^{21}$Ne concentrations in the majority of quartz clasts are in excess of $10^9$ atoms/g (Supplementary Data 1, Supplementary Information Fig. S6). Such high concentrations integrate the long-term exposure of the quartz clasts to cosmic rays at the surface (<3 m[50]), including initial exhumation from the source bedrock and subsequent transport to, and deposition at, the final site. Uplift-corrected $^{21}$Ne exposure ages from all sites (Supplementary Data 1) range from 62.5 to 1.2 Ma (Fig. 2). Thirty-two clasts (24 %) yield modelled exposure ages that are older than the Oligocene-Miocene boundary, the earliest estimate of the onset of hyperaridity in Atacama Desert so far[15–17]. The abundance of old clasts is remarkable given that long-lived clasts are least likely to be preserved on surfaces. We additionally analysed seven samples for their $^{10}$Be concentration with all being at or near saturation (Supplementary Data 1), confirming that the clasts have been at the surface for at least ~5 Myr (since the Late Pliocene) with minimal to no burial during this period[50]. Near-saturated to saturated $^{10}$Be concentrations, together with high $^{21}$Ne inventories, imply negligible or exceptionally low net clast erosion over at least the past few million years. Saturation or secular equilibrium of $^{10}$Be requires that samples have remained at the surface, or surface lowering has been minimal over timescales of the past millions of years (~4–5 Myr, the characteristic time to reach secular equilibrium).

The quartz clasts from the Jazpampa surfaces were deposited after the eruption and deposition of the tephra exposed in Quebrada Tiliviche. New U–Pb zircon ages of the lowermost and uppermost tephra layers are $25.17 \pm 0.48$ Ma (n = 4) and $19.01 \pm 0.83$ Ma (n = 4) respectively (Supplementary Data 2). Younger age limits the earliest time at which clasts could have been deposited on the overlying low-relief surface. The majority of clasts, however, yield modelled apparent exposure ages that are significantly older than the time of surface formation. Consequently, the clasts must have acquired cosmogenic Ne prior to final deposition during slow exhumation and transport. Thus, to avoid confusion between clast deposition age and the time the clast has been at the Earth's surface, we use the term exposure duration when referring to the measured concentrations of cosmogenic $^{21}$Ne in quartz clasts. This more accurately represents the total time a clast has been exposed to cosmic rays, and includes exhumation, transport, shallow burial and final deposition. This approach acknowledges that cosmogenic $^{21}$Ne concentrations reflect the integrated production history rather than a single, continuous episode of exposure at their current positions. Thus, the very long exposure durations recorded by the clasts from the Jazpampa surfaces imply that the region preserves a record of very slow landscape development consistent with the onset of intense aridification significantly earlier than the Early to Mid-Miocene[15,16,19,27]. Under such conditions, using the measured cosmogenic nuclide data to calculate quantitative erosion rates is not appropriate, as steady-state assumptions are violated and nuclide inventories gained during erosion from bedrock cannot be separated from post-erosion production.

### Evidence for Pre-miocene hyperaridity
Before exploring the implications of quartz clasts with pre-Miocene exposure durations, it is crucial to rule out alternative explanations for the high $^{21}$Ne concentrations. Non-cosmogenic sources are not a viable

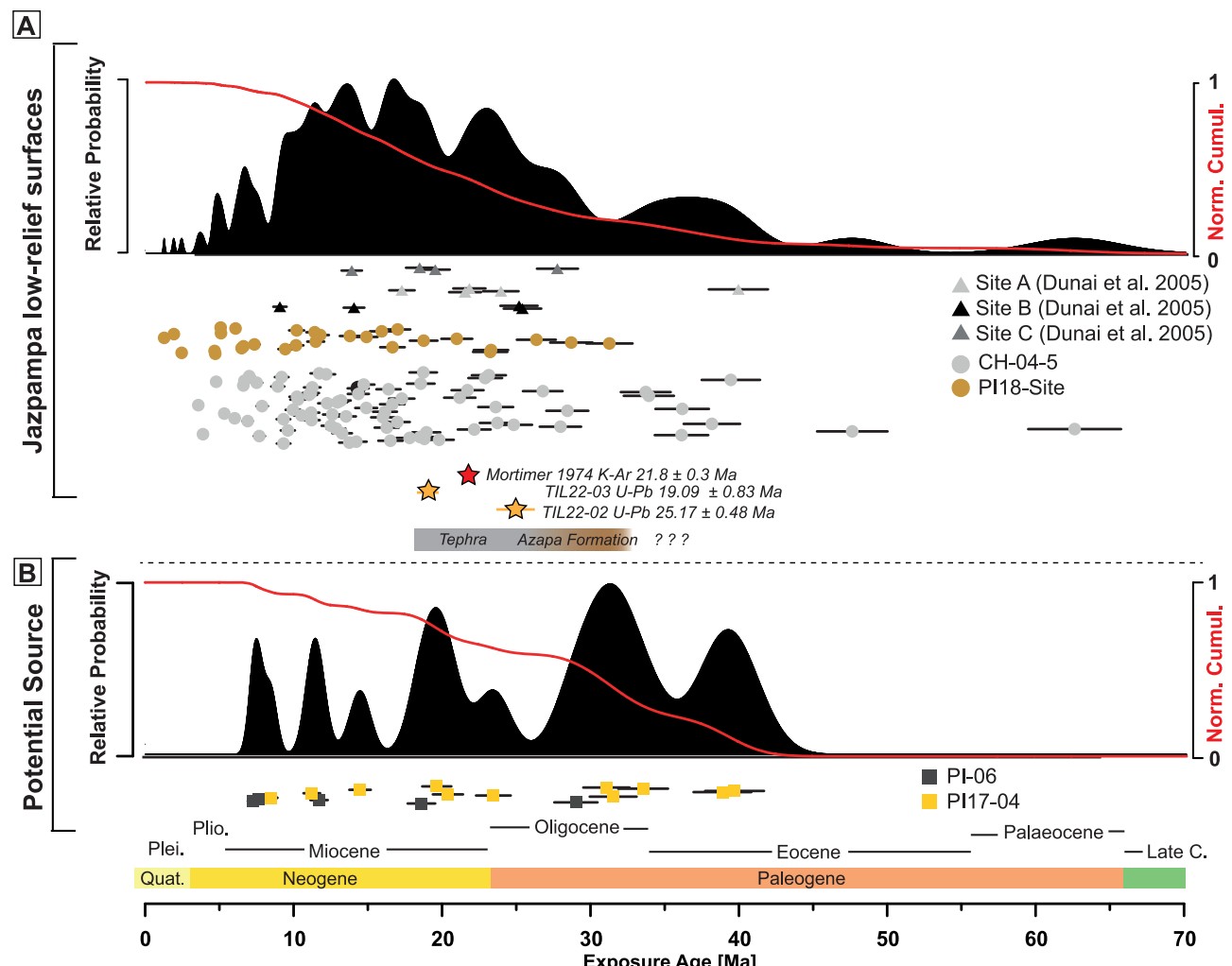

**Fig. 2 | Detailed Chronological Data.** Results $^{21}$Ne exposure ages and tephra U-Pb zircon ages. Probability density plots of single-clast $^{21}$Ne uplift-corrected exposure ages/durations, including individual ages ($\pm 1\sigma$), from **A** the depositional surface and **B** from the catchment and equivalent surface to the south. The red curves indicate the normalised cumulative curves of the exposure histories. The cosmogenic nuclide data have been recalculated from Dunai et al.[15]. Stars indicate tephra ages. The brown bar indicates the potential age range for the deposition of the Azapa Formation at our study site. The grey bar indicates the range of ages for the tephra deposits at our study site.

explanation (Supplementary Information Fig. S6). Rarely nucleogenic neon mimic cosmogenic neon isotopically[51]. In this case it can be completely excluded as the low U and Th concentrations in these quartz samples (<6 ppb; Dunai, et al.[15] supplementary data) produce several orders of magnitude less Ne than measured in these samples. Previous studies have suggested that inherited cosmogenic Ne might explain high cosmogenic nuclide concentrations[19,52]. The only reasonable explanation by which the vein quartz fragments could have acquired high concentrations of cosmogenic $^{21}$Ne, other than by long exposure in an achingly slow evolving Coastal Cordillera landscape, is by shorter exposure at higher elevation. High altitude Andes-derived sediments could only have been delivered to the region prior to the Late Oligocene-Early Miocene when the Coastal Cordillera became isolated[15,33–35]. Prior to isolation, the Western Cordillera did not exceed ~2.5 km elevation[53,54] (Fig. 2C). If the old clasts were derived from the Western Cordillera, the cosmogenic $^{21}$Ne in excess of what was acquired after deposition at 23 Ma, would require exposure up to ~7 million years at the surface (Supplementary Data 1). Such long exposure durations require very low erosion rates (<0.1 m/Myr, Supplementary Data 1), which in turn implies very low rates of landscape change prior to the Miocene-Oligocene boundary in the uplifting Andes (and would be indicative of pre-Miocene landscape stasis).

However, such low erosion rates for the emerging Western Cordillera are highly unlikely[25,53]. This is confirmed by low cosmogenic $^{21}$Ne concentrations measured by Dunai, et al.[15] in rounded quartz clasts from the Azapa Formation fluvial gravels that underlie the Jazpampa surfaces (Fig. 1, Site D in ref. 15, Supplementary Information Fig. S2). Two samples of amalgamated clasts (25 clasts each) yield cosmogenic $^{21}$Ne concentrations equivalent to less than 140 kyr of exposure (Supplementary Data 1). The absence of pre-Miocene exposure ages in clasts from Precordillera alluvial fans[17] is a further argument that the Andes have never shed long-exposed sediment. With the exception of two exceptionally long-lived clasts (CGN-CH04-5-14 (~ 47 Myr) and CH04-5-72 (~ 62 Myr)), the majority of the old clasts have been at the surface for 20-40 million years (Fig. 2). The abundance of clasts with Mid to Late Eocene exposure durations (Fig. 3) implies that the decline in the strength and intensity of fluvial periods in the current hyperarid core occurred significantly earlier than previously considered, around Mid to Late Eocene. Despite the possibility that intermittent burial of clasts may mean that the modelled exposure times are likely lower limits, there is a remarkable similarity of the distribution of quartz clast exposure ages with the supergene mineralisation age (Fig. 3B). This further strengthens the need to reconsider the aridification history of the Atacama Desert.

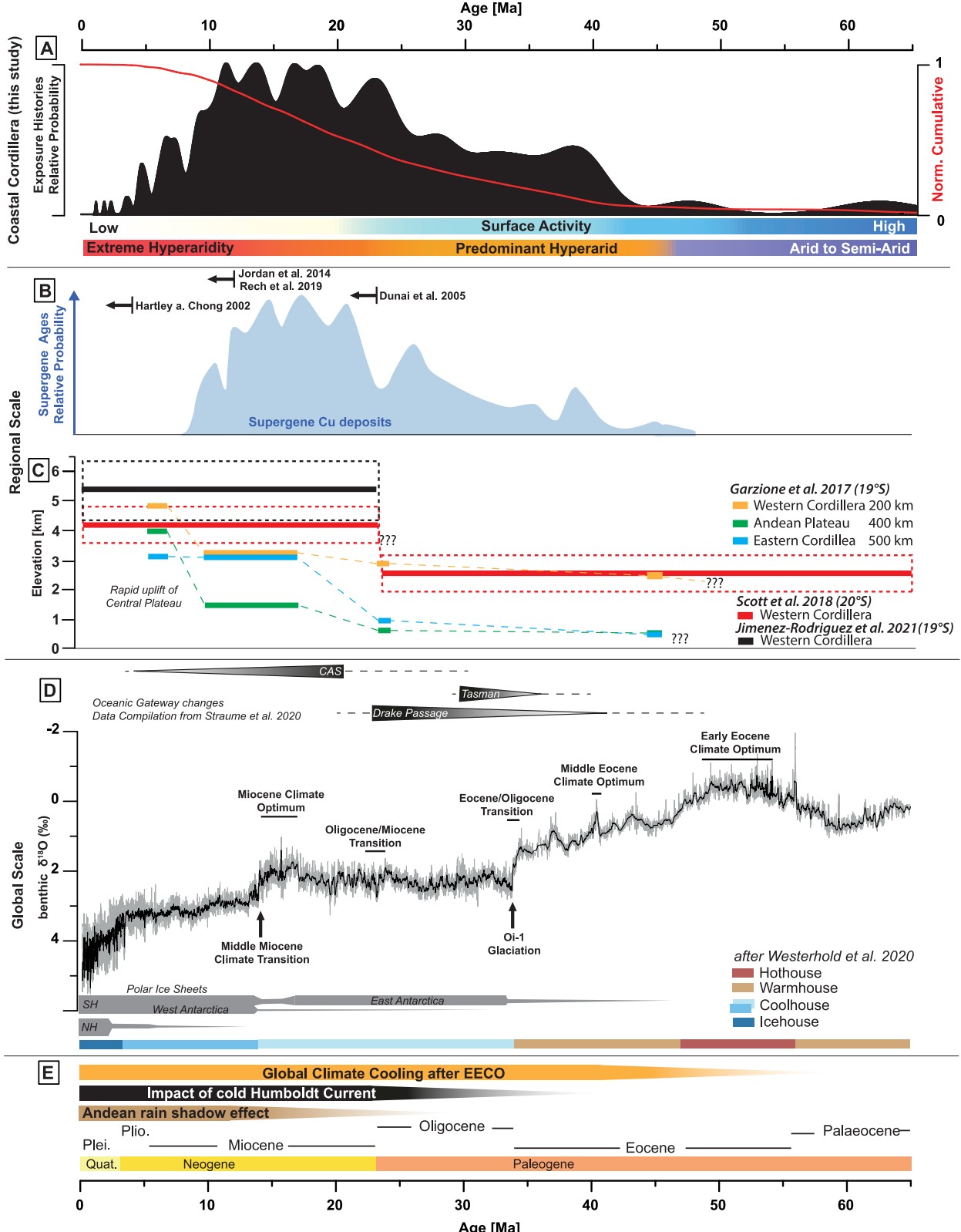

Younger ages (<23 Ma) are consistent with the notion that intermittent brief periods of wetter conditions and regional tectonic activity interrupted the predominant hyperarid climate[15–17,19,22]. The low number of clasts with exposure ages of less than 10 Ma is consistent with the establishment of hyperarid conditions during the Mid-Miocene[19,27,55] throughout the entire Atacama Desert (including areas in the Precordillera) and intensification towards extreme hyperarid conditions in the core of the Atacama Desert.

The prolonged and predominantly hyperarid climate of the Atacama Desert promotes a positive feedback mechanism by enabling the accumulation and long-term preservation of atmospherically deposited gypsum dust, a process referred to as soil inflation and ubiquitous across gypsisols of the region[37–39,56]. Widespread gypsum surface

**Fig. 3 | Compilation of ²¹Ne exposure ages and relevant regional and global paleo-datasets. A** Relative probability plot of all new and previously published modelled cosmogenic Ne ages of clasts from the Jazpampa surfaces. Bars indicate qualitative reconstruction of surface activity and aridity state based on our data. **B** Compilation of supergene mineralisation ³⁹Ar/⁴⁰Ar ages from the currently arid Precordillera and Western Cordillera of the Atacama Desert from Reich and Bao[78] and proposed onsets of hyperaridity, shown by the black left-pointing arrows with citations at the top of the figure panel. **C** Andean uplift reconstructions after Garzione, et al.[53] for the Western and Eastern Cordillera and for the Andean Plateau (taken from Fig. 4 (central CAP 16–19°S) at 200, 400 and 500 km profile lengths) and uplift reconstructions for the Western Cordillera (20°S) from Scott et al.[54] and from Jimenez-Rodriguez et al.[79]. **D** Oceanic gateway changes during the Neogene according to data compiled by Straume et al.[6]. Global Cenozoic reference benthic foraminifer oxygen isotope dataset (CENOGRID) from Westerhold et al.[5] covering the last 65 million years. The vertical grey bars show a qualitative representation of the ice volume in each hemisphere according to Westerhold et al.[5]. Major global climate periods/events are marked, and climate states are defined by Westerhold et al.[5]. **E** Overview of the key potential drivers influencing the increasing aridity in the hyperarid core of the Atacama Desert. Global climate cooling since the EECO based on the Global Cenozoic reference benthic foraminifer oxygen isotope dataset (CENOGRID) from Westerhold et al.[5]. Increasing impact of the cold Humboldt Current with the gradual opening and deepening of the Drake Passage based on oceanic gateway changes during the Neogene according to data compiled by Straume et al.[6] and increasing rain-shadow effect of the uplifting Andes based on data compilation from (**B**), with full impact of the recent rain-shadow effect by the uplifting Altiplano-Puna Plateau since the Mid Miocene.

covers derived from atmospheric dry deposition[39] have been documented and are subsequently modified under limited moisture availability, such as rare precipitation events, fog, or dew[57–59]. These surfaces undergo induration through encrustation and cementation, which enhances sediment strength, armours the landscape, reduces overland flow, and protects surfaces from fluvial reworking during episodic rainfall[22]. "Born-at-the-surface" models further explain the persistence of pavement clasts while gypsum soils continue to accumulate beneath them[60,61]. Together, these processes increase the critical precipitation threshold required to trigger surface activity and erosion[22], thereby promoting long-term surface stability and supporting long clast exposure durations, consistent with cosmogenic ²¹Ne concentrations and the interpretation of predominant hyperarid conditions over million-year timescales in parts of the Atacama Desert.

## Potential causal links between aridification in the Atacama Desert and the global climate

Despite the potential for the uplift of the Andes and the development of the Humboldt Current to reduce regional precipitation, we hypothesise that the current dry core of the Atacama Desert has already experienced extreme aridity long before the early Miocene. Aridification driven by the 'orogenic–orographic feedback effect' described by Evenstar et al.[24] post-dates the onset of hyperarid conditions inferred from long-lived clasts in the Coastal Cordillera. This suggests that, while Andean uplift may have driven the progressive eastward expansion of aridity into the Precordillera, Western Cordillera and Altiplano Plateau, the establishment of predominantly hyperarid conditions in the Coastal Cordillera likely occurred significantly earlier.

The new data tend to rule out the severe global cooling at the Eocene-Oligocene boundary related to the Oi-1 glaciation[5,62] as the trigger for intense aridification of the Chilean Coastal Cordillera (Fig. 3D). The preservation of clasts with exceptionally high cosmogenic Ne concentrations implies that extreme landscape stability initiated already during the Eocene. Given that the modelled exposure durations may underestimate the true exposure history of the pebbles, it is possible that global climate cooling following the Early Eocene Climate Optimum (EECO)[5] was the trigger (Fig. 3A, D, E). An Early to Mid-Eocene aridification would require that the regional climate system passed a threshold after the EECO, but prior to the opening of the Drake Passage and subsequent formation of the Antarctic Circumpolar Current (ACC, see Fig. 3E). It is conceivable that the upwelling of a cooler proto-Humboldt Current, with water masses originating from high latitudes, may have already exerted an influence on the climate of the west coast of South America since the Late Cretaceous[63]. Keller et al.[63] proposed that prior to the opening of the Drake Passage and the subsequent establishment of the Antarctic Circumpolar Current during the Eocene/Oligocene (Fig. 3D)[6], the intensity of the proto-Humboldt Current may have been significantly stronger due to the focused northwards flow of high-latitude waters. Although these high-latitude waters were likely warmer than modern water masses, the more intense proto-Humboldt Current could have cooled surface waters off the Chilean coast, ultimately leading to the progressive aridification of the Atacama Desert.

## Methods

We used geological field observations and a digital elevation model (DEM) of the study area (Advanced Spaceborne Thermal Emission and Reflection Radiometer (ASTER) Global DEM (GDEM) V3, ~30m resolution) to delineate drainage catchments. Satellite imagery (Earthstar Geographics SIO, ©2023 MAXAR) was additionally used to identify geomorphological features.

### TCN Analysis

The chronology based on exposure times was established using ²¹Ne and ¹⁰Be cosmogenic nuclide exposure dating. Cosmogenic ²¹Ne exposure ages were determined for quartz clasts sampled during field expeditions in 2004, 2017, and 2018 (Supplementary Information Fig. S1). The quartz clasts were sampled from two different areas: (1) the sources of eroded sediment, i.e., the headwaters of the studied catchment (samples PI06-1-4 and PI17-004); and (2) the low-relief depositional Jazpampa surfaces downstream (samples CH04-5, PI18-001, PI-03, PI-06, PI-07, PI-11, and PI-12). Details of each individual sampling site are provided in the Supplementary Information Figs. S2 and S3, and have been published in ref. 15.

Samples were crushed, sieved to retain the 250-710 μm grain-size fraction and etched several times in a dilute HF-HNO₃ mixture[64]. Purified quartz separates were investigated under a microscope. Etched samples were used for ¹⁰Be and ²¹Ne analyses.

Etched quartz samples were dissolved in concentrated HF/HNO₃ after spiking with a certified Be standard solution (Supplementary Data 1). Accelerator mass spectrometry (AMS) target preparation followed the standard procedure described in ref. 65 with minor modifications. Following calcination to BeO, samples were mixed with Nb powder prior to pressing. Chemical blanks were prepared in parallel with the samples. ¹⁰Be/⁹Be values were measured at the Scottish Universities Environmental Research Centre (SUERC) AMS Laboratory. ¹⁰Be/⁹Be ratios were normalised to NIST SRM 4325 with nominal ¹⁰Be/⁹Be = 3.06E-11[66]. Blank-corrected concentrations of ¹⁰Be were derived following the procedure described in ref. 67. Concentration uncertainties include the propagated uncertainties in the AMS ratios of both the sample and the appropriate blank, together with the estimated uncertainty in the ⁹Be masses added to the samples and blanks during spiking.

²¹Ne samples were measured at Vrije Universiteit (VU) Amsterdam (site CH04-5, dataset of[15]), at SUERC (site PI06-1-4)[68] and at the University of Cologne (UoC), (PI18-001, PI17-004)[69] using noble gas mass spectrometers. Samples CH04/5 1-80 and samples published in ref. 15 were heated to 1000 °C for 15 minutes, purified and subsequently measured on the noble gas mass spectrometer at the VU Amsterdam[70]. The uncertainties of the cosmogenic ²¹Ne concentration of samples CH04/05 1 - 80 reflect the average analytical reproducibility of ²¹Ne determinations of 2%, at the time of analysis (until August 2005) at the

VU-Amsterdam. 10 splits of CH04-5 samples were re-measured at the Cologne noble gas mass spectrometer. Based on the observed average offset (-5.55% for $^{21}Ne_{excess}$) all results from Amsterdam were corrected. At the time of analysis in Amsterdam (i.e., prior to 2005), no international intercalibration material was available. With our correction, the legacy measurements are related to the CREU1 intercomparison material and current measurements at the UoC[69,71]. The correction included the mean uncertainty of the calibration gas measurements during the analysis period (0.8%). Further information in Supplementary Data 1.

Purified samples (PI-06) were packed into aluminium foil cups and were measured with the noble-gas mass spectrometer at SUERC applying the standard procedure, including correction for isobaric interferences at masses 20, 21, and 22[68,72]. Additional samples from recent field campaigns (PI18-001, PI17-004) were measured on the noble gas mass spectrometer at the UoC using the analytical methods described in ref. [69]. CREU quartz standards were measured at SUERC and UoC for interlaboratory comparability and quality control[71]. The spallogenic origin of the measured $^{21}Ne$ excess was verified using the triple isotope plot (Supplementary Information Fig. S6, Supplementary Data 1).

### TCN Calculation

The $^{21}Ne$ exposure ages were calculated using SPRITE[73], a model framework for scaling cosmogenic nuclide production rates of pre-Quaternary timescales. We applied an uplift rate of 40 m/Myr (see[15] and Supplementary Data 1). SPRITE scaling stops the uplift rate correction once one sample of the entire data package reaches sea level. We calculated the exposure age uncertainty by subtracting/adding the concentration (lab) uncertainty to the measured concentration and calculated the minimum and maximum exposure ages. From the later we calculated the minimum and maximum uncertainty, which mainly reflect only the lab uncertainty. To account for the uncertainty regarding the scaling and the uplift rate (geological uncertainty), we assumed a total of 5% for all samples.

Pre-exposure of amalgamated quartz clast samples from[15] (Site D) were re-calculated using Cronus Earth Calculator with the sample site data from[15], assuming no uplift correction. Approximation of potential pre-exposure prior to 23 Ma in high altitude catchment in the uplifting Andes were also calculated with the Cronus Earth Calculator, using 2.5 km altitude and GPS position of a site east of our study area within the Andes. We applied no uplift corrections. All variables and calculated results are deposited in the Supplementary Data 1.

$^{10}Be$ ages were calculated using the LSDn scaling scheme of[74], as implemented in version 3 of 'the online calculators formerly known as the CRONUS-Earth online calculators' (https://hess.ess.washington.edu/math/v3/v3_age_in.html)[75]. We applied the uplift correction as stated in ref. [22] using an uplift rate of 40 m/Ma. All $^{10}Be$ samples have concentrations at or close to their saturation limit/secular equilibrium (<4-5 Ma) and can therefore be considered as the minimum exposure age/duration of the clasts. Further information is in the Supplementary Data 1.

### U-Pb LA-ICP-MS analysis

Tephra samples were prepared using standard mineral separation techniques at the UoC. Isotope mass spectrometry was carried out at the Institute of Geosciences at Goethe University Frankfurt. Hand-picked zircon grains were mounted in a 25 mm diameter circular epoxy mount and polished to approximately half of their thickness. The internal structure of the zircon crystals was investigated using cathodoluminescence imaging. Zircon U-Pb isotope analysis was conducted with laser ablation inductively coupled plasma mass spectrometry (LA-ICP-MS) using a ThermoScientific ElementXr sector field ICP-MS coupled to a Resolution M-50 (Resonetics) 193 nm ArF

excimer laser (CompexPro 102, Coherent). Technical details can be found in the Supplementary Data 2.

Tephra samples were measured in two different analytical sequences, TIL-02 in sequence 1 and TIL-03 in sequence 2. Ten U–Th–Pb spot analyses were carried out on sample TIL-02 and 16 on sample TIL-03. Due to the small number of crystals recovered from these samples, all zircon crystals were analysed. Maximum depositional ages (MDAs) for each sample were calculated following the most conservative method (YC2σ (3 + )) reported by ref. [76]. MDA calculated for sample TIL-02 is 25.17 ± 0.48 Ma (n = 4) and MDA for sample TIL-03 is 19.09 ± 0.83 Ma (n = 4). U–Th–Pb data and uncertainties (2S) are reported in the Supplementary Data 2, see also Supplementary Information Fig. S7.

**Plotting of TCN data in Figs. 2 and 3.** We used the 'fancy-pants' plot scheme (modified normal kernel density) to plot our data, which have a wide range of exposure ages/durations (https://cosmognosis.wordpress.com/2018/09/25/the-fancy-pants-camelplot/). We use a 5% uncertainty for our exposure age/duration data to adjust the calculated kernel height.

## Data availability

All the data generated or analysed during this study are included in this published article (and its Supplementary Information files). Supplementary Data 1 Supplementary Data 2.

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

## Acknowledgements

We want to thank Eduardo Campos, Gabriel González López, and colleagues from the Universidad Catolica del Norte at Antofagasta as well as Sheng Xu at the SUERC AMS Laboratory, and Luigia DiNicola at the SUERC Noble Gas Laboratory for their support. Additional we want to thank Moe Mijjum for her help using SPRITE.

## Author contributions

B.R. contributed to fieldwork, sample preparation, noble gas isotope analysis, data evaluation, and manuscript writing. S.A.B. contributed to fieldwork. SAB and DF. contributed to [10]Be analyses and data evaluation. F.A.S. contributed to noble gas isotope analysis, data evaluation and manuscript writing. R.A. contributed to U–Pb dating. VW contributed to the tephra sample preparation. T.J.D. contributed to fieldwork, sample preparation, neon isotope analysis, data evaluation, and manuscript writing. B.R., S.A.B., F.A.S., D.F., R.A., V.W., and T.J.D. reviewed the manuscript.

## Funding

This project is affiliated with the Collaborative Research Center (CRC) 1211 and funded by the German Science Foundation (DFG), Projektnummer 268236062. Open Access funding enabled and organized by Projekt DEAL.

## Competing interests

The authors declare no competing interests.
