## [Peer Review file · Nature Communications]

Evidence for Eocene aridification of the Atacama Desert's hyperarid core

Corresponding Author: Dr Benedikt Ritter-Prinz

Version 0:

Reviewer comments:

Reviewer #1

(Remarks to the Author)

Review of Ritter-Prinz et al., "Eocene aridification of the Atacama Deserts' hyperarid core."

Prepared by Marissa Tremblay, 2025.07.07

This manuscript by Ritter-Prinz and coauthors presents new cosmogenic ^{21}Ne measurements from quartz clasts exposed in the low-relief alluvial surfaces the Chilean Coastal Cordillera as well as new U-Pb ages of tephra deposits located beneath these surfaces. Combined with previous cosmogenic ^{21}Ne measurements and other observations from the area, the authors argue that the onset of hyperarid conditions in this part of the Atacama Desert began in the Eocene, which is significantly earlier than the previously suggested Miocene onset. The authors hypothesize that Eocene hyperaridity could be linked to cooling after the Early Eocene Climatic Optimum and a stronger proto-Humboldt Current.

Overall, I found this paper to be well written, and the presentation of integrated exposure ^{21}Ne ages as old as the Eocene is an exciting result. I believe this is a valuable contribution that challenges our current understanding of the onset of hyperarid conditions in the Atacama, and it will inspire additional research on what the mechanisms for such an early onset of hyperaridity might be. My comments on the manuscript are relatively minor and should be straightforward to address. I recommend publication following minor revisions.

General comments:

Uplift scenarios: The uplift scenarios tested, as shown in Datable_05_Uplift Models, all have the sample sites reaching their current elevations yesterday. An uplift scenario like this will tend to yield older exposure ages, simply because the samples will spend most of their exposure histories at lower elevations where the production rates are lower. I am not suggesting that these uplift models are incorrect, but it would be helpful if the authors could discuss why uplift of the Coastal Cordillera could not have occurred earlier, such that the surfaces they studied sat near present-day elevations for some prolonged period of time (~Myr). For example, citing geodetic observations or observations from coastal terrace uplift to support the argument that the Coastal Cordillera is experiencing present-day/recent uplift would be helpful.

Data presentation: I think Figure 1 would be much stronger if there were a panel with the individual clast cosmogenic ^{21}Ne ages included, rather than only showing the individual ages in Extended Data Fig. 6A. If the figure is limited in space, I recommend removing the oxygen isotope curve and simply including the climatic events referred to in the text (EECO, Oi-1 glaciation) with the bar charts for the ice sheet chronology that appears beneath the O isotope curve.

Does the relative probability plot in figure 1A include both the ^{21}Ne ages from the depositional surface and from the catchment to the south (i.e., the separate relative probability plots in 6A and 6B)? If so it might be helpful to state that in the caption for Extended Data Fig. 6, as I was only comparing Figure 1 and Extended Data Fig. 6A originally and was confused about where the subtle differences between the two were coming from.

The authors should include an extended data figure that presents their new tephra U-Pb data (probably the compilation of U-Pb age PDFs, with the MDA annotated), rather than having these data only appear in the supplementary Excel spreadsheet.

Finally, I think the authors should add the nucleogenic-air mixing line to Extended Data Fig. 7.

Line-by-line comments:

Line 22: Later in the manuscript on line 59, the authors state they have measured ^{21}Ne in only 122 quartz clasts. Which is correct?

Line 50: The word 'the' is needed before 'Coastal Cordillera'.

Line 174: I don't think it's a fair statement to say there is a near absence of clasts with exposure ages less than 10 Ma. From `Datatable_04_Ne_Ages` in the Excel spreadsheet provided, I count 34 clasts with exposure ages less than or equal to 10 Ma. For comparison, there are only 9 clasts with exposure ages in the data table that are greater than or equal to 34 Ma (i.e., Eocene or older).

Line 324: Is this fault scarp shown in the satellite imagery (I'm fairly certain it is)? If so, I recommended annotating the imagery with its location, like how you've noted the location of the Quedabra de Tiliviche.

Reviewer #2

(Remarks to the Author)

This is an interesting dataset that is worthy of publication but may be in a longer format paper where a more detailed discussion of the dataset can be presented and the interpretations outlined more clearly.

Introduction should be written with a clear narrative – at the moment it jumps around from present day to rock record, and general to detailed from paragraph to paragraph. The information is there but needs to follow more coherently.

L39 remove 'Cenozoic' as record is back to 150 Ma

Some clarity regarding the exact geomorphic relationships would be good. The sampled quartz clasts occur on and above dated tephra and the Azapa Formation of late Oligocene-Miocene age. Could the authors clarify how the exposure ages of the quartz clasts are older than the sediments on which they occur? If, as noted, the area is hyper-arid it is unlikely that the quartz clasts have been transported any significant distance so how did they end up on the Azapa/tephra? A model illustrating how the quartz clasts arrived at their final resting place would be useful to explain to the reader. Is the argument that they have been exposed in the Coastal Cordillera since the Eocene and were then periodically reworked onto the Azapa/tephra surface but retained their exposure age? Transport distance of 5 km or more are suggested in Fig. 1. Some comment on how much reworking/burial would need to take place to change the exposure age would be useful.

Sample PI-17 is not in the catchment to the main sample set and it's not easy to get a drainage system to link to the sample set from PI-06 either. Given that there has been little or no erosion since the Eocene drainage systems should be broadly similar (with the exception of the NE-SW faults which are younger), such that it is not easy to clearly link the samples in the catchments to those on the tephra/Azapa surface.

The conglomerates of the Azapa Formation clearly indicate that it was wetter at times in the study area in the Oligo-Miocene. How do the authors reconcile the deposition of the Azapa Formation in the Miocene with hyperaridity from the Eocene onwards.

The fact that supergene ages also extend back into the Eocene (Fig. 2) indicates that a hyper-arid climate was in place in areas of the Atacama since that time period, so the fact that there are clasts with an Eocene age of exposure is not unexpected, but the fact that they have been preserved in some areas is definitely of interest.

A possible origin for onset of Eocene hyperaridity could be related to eastward expansion of the Central Andes. Uplift in the Subandean Zone/Eastern Cordillera is the principal orographic control in the Central Andes and was most likely initiated around 50 Ma (see Evenstar et al 2023 EPSL for more details).

In conclusion this is a very interesting manuscript which raises some fascinating questions about the history of hyper-aridity in the Atacama Desert. However, without further documentation and explanation it is not clear how the clasts have retained their exposure ages whilst being transposed onto a Miocene surface, and needs to be more clearly explained/documented. The transport of clasts significant distances without exposure ages being reset needs to be explained. The fact that hyperaridity in the core of the Atacama is already known to extend back to the Eocene based on supergene enrichment ages indicates that the implications are not as great as suggested.

Reviewer #3

(Remarks to the Author)

This is a well-written paper presenting new exposure ages to support an early hyper-aridification of the Atacama Desert. The introduction clearly explains the global importance of this topic. This contribution builds on the Dunai et al. (2005) Geology manuscript and extends those findings to suggest an earlier onset of hyper-aridity. Like the Dunai et al. paper, this paper uses cosmogenic ^{21}Ne to model surface exposure ages of clasts and uses these dates to constrain the timing of hyper-

aridity. And, like the Dunai et al. paper, this one suffers from the uncertainty of the role of ^{21}Ne inheritance in the measured concentrations of the clasts. Namely, that it's impossible to know how much of the ^{21}Ne was produced in the clasts in the sampled surface exposure versus how much was produced in the rock at some point in its near-surface exposure history. This paper addresses this uncertainty directly, but can offer only suggestive explanations for why their modeled ages are in fact surface exposure ages without any inheritance. They also use an incomplete and carefully selected set of references, rather than a comprehensive set to support their suggestions. Their suggestive (and speculative) explanation, coupled with the correlation with supergene enrichment ages leads the authors to forward their results in a definitive, rather than speculative, way. There are simply too many of the same flaws that were articulated in rebuttals of the Dunai et al. (2005) findings in this paper.

The details of these explanations, as well as untangling the methodology used lend themselves better to a specialty journal than a high profile, broad audience journal such as Nature. These results deserve to be published and put into the scientific discourse such that they can be more fully debated and assessed. For this to happen, the details of the sample collection and analyses would need to be fully fleshed out in a longer and more detailed paper. Similarly, an articulation of a more thorough explanation of how these results complement and build on the Dunai et al. (2005) data would need to happen as this paper often reads like a repeat of that paper. Unfortunately, I cannot recommend publishing the paper in Nature.

Version 1:

Reviewer comments:

Reviewer #1

(Remarks to the Author)

Review of Ritter-Prinz et al., "Eocene aridification of the Atacama Deserts' hyperarid core."

Prepared by Marissa Tremblay, 2026.01.23

My comments on the original manuscript by Ritter-Prinz and coauthors were minor, and I find the authors have done a sufficient job answering my queries and critiques. I also think the authors have done a good job of responding to the comments of the other two reviewers and revising the manuscript accordingly. I also found reviewer #3's criticisms of the original manuscript to be vague and therefore difficult to address. I only have minor follow up comments regarding the revised version:

Lines 18, 23, 55: To be consistent in your revisions, I recommend changing the phrasing in these places from "exposure ages" to "exposure durations" as well.

Line 108: You may consider revising to "rare, localised torrential rain events."

Line 146-147: You may consider revising to "modelled apparent exposure ages."

Line 212: Given the acknowledged uncertainties regarding the uplift history, I recommend rephrasing from "probably underestimate the true exposure history" to "may underestimate the true exposure history."

Reviewer #4

(Remarks to the Author)

Ritter-Prinz et al. present a new dataset of exposure ages derived from clasts on low-relief alluvial surfaces in the Coastal Cordillera. They argue that many clasts record mid- to late-Eocene exposure durations, with implications for how low landscape-evolution rates may be. The results also push the onset of aridity back to well before the classic early Miocene Humboldt Current/Andean uplift explanation. I read the authors' response to the reviewers, and I think they have either addressed or plausibly responded to the raised concerns. The idea is creative, and I therefore recommend publication after addressing the following comments.

I comment on this manuscript primarily from a geomorphology and landscape-evolution perspective. I do not have specific expertise in cosmogenic-nuclide exposure dating, so my comments focus on the geomorphic interpretation, assumptions, and the clarity of the landscape-evolution argument rather than the technical details of the nuclide methods.

1) The manuscript uses qualitative phrasing (e.g., "very slow landscape evolution," "long exposure durations") to link the exposure-age results to hyper-arid conditions, but it does not clearly quantify how slow the landscape evolution would need to be. It would strengthen the argument to translate these qualitative statements into quantitative bounds. For example: how much net erosion and/or deposition could have occurred while still preserving the inferred exposure durations and the observed surface/clast characteristics?

Is there a way to benchmark the implied rates against modern analogs in hyperarid to arid deserts (or even semi-arid settings) to show whether the required erosion/deposition rates are uniquely consistent with hyperaridity? In other words, could the inferred low long-term rates instead reflect a system dominated by a small number of high-magnitude events superimposed on otherwise low background geomorphic activity, rather than requiring persistently hyperarid conditions?

If there is no way to such quantification, I encourage the authors to acknowledge this limitation as it could be arid, semiarid climate with a small number of events with high-magnitude can do same geomorphic work, low rates of landscape evolution.

2) Null hypothesis: I think the authors should evaluate whether non-climatic processes such as clast recycling, intermittent burial and exhumation, and surface armoring could also explain the observed exposure ages.

3) Burial and exhumation: Could the authors conduct sensitivity tests to quantify how much intermittent burial (in terms of depth and duration) could occur while still reproducing the observed exposure age distribution?

Overall, I think the authors have already addressed the previous reviewers' comments, and the manuscript is an important contribution to understanding the onset of aridity in one of the world's most important deserts. My comments are intended only to improve the geomorphic aspects of the paper, or at least to encourage the authors to acknowledge the study's limitations.

Best wishes,
Abdallah Zaki

Reviewer #5

(Remarks to the Author)

Review of Ritter-Prinz et al., "Eocene aridification of the Atacama Deserts' hyperarid core."

This review is based on the revised manuscript, taking into account the comments and refutations of three previous reviewers.

The article hypothesizes that the onset of aridity/hyperaridity in the Atacama Desert may have occurred much earlier (since the Eocene) than previously established (Oligocene-Miocene boundary). This chronological shift is based on the study of accumulated cosmogenic signals analyzed in clasts deposited on low-gradient, minimally eroded surfaces in one of the most arid sectors of the modern Atacama Desert. Most of the data are derived from a previously published study (Dunai et al., 2005). Once presenting their hypothesis, the authors propose that the primary driver of Eocene aridity was the global cooling that occurred prior to the Eocene Climate Optimum.

The available data are insufficient to present the hypothesis as an unequivocal thesis; it remains a weak proposal. From this reviewer's perspective, there are significant stratigraphic and sedimentological arguments that hinder the acceptance of the hypothesis and even render it somewhat contradictory:

1. Stratigraphic Constraints: The deposition of the analyzed gravels on flat, uneroded surfaces certainly occurred after 12 and 25 Ma (Oligocene-Miocene boundary to Miocene), as evidenced by the age of the tephra underlying the gravels. This a priority and definitively constrains the age of both the deposit and the planation surface.

2. Transport Dynamics: Due to model requirements, the authors indicate that the spatial transport of clasts to the depositional area was short. It is stated that the materials deposited in the Miocene were transported within relatively closed and local sectors of the Coastal Range, isolated from inputs from the Precordillera and the Andes. This short-distance transport suggests it could not have been extended in time. Consequently, it is difficult to reconcile a ~25 Ma interval (the duration between the Eocene and the lower Miocene boundary) to complete the transport from a hypothetical Eocene exhumation zone to the Miocene depositional site.

3. Lack of Eocene Surfaces: The authors have not found—nor do they mention—flat/uneroded surfaces containing Eocene-aged deposits with clasts exposed since that period. Such a finding would constitute unequivocal evidence of Eocene hyperaridity. In fact, the absence of these deposits and surfaces suggests that, if they did form, they were likely eroded during a humid, non-arid Eocene period.

I consider that is the Editor's decision whether to accept a proposal based on the data provided. In its current form, it constitutes solely a working hypothesis. To be definitively proven, the authors' idea requires additional study sites in the region, including the discovery and analysis of Eocene deposits and surfaces. Should it be published, the authors should adopt a more tentative tone in both the title and the abstract. A more suitable title would be: "Indications of the transition suggesting arid conditions in the Atacama Desert during the Eocene."

Beyond these primary considerations, the authors could improve the explanation of the causes for the proposed Eocene hyperaridity. This could be achieved by adding synoptic maps of the region illustrating the regional factors driving aridity, with separate maps for the Eocene and Miocene stages. Furthermore, the manuscript would be enriched by a discussion on the emergence, existence, and variability of SE and N-NE summer monsoons as mechanisms for moisture/precipitation delivery to Atacama latitudes, as demonstrated Sáez et al. (2016) for the Holocene Sáez et al.

Version 2:

Reviewer comments:

Reviewer #4

(Remarks to the Author)

I have read the revised manuscript and the authors' responses to the comments raised by me and other reviewers. I appreciate the efforts made to address the concerns, and I think the manuscript is ready for publication. Thank you.

Reviewer #5

(Remarks to the Author)

This reviewer has carefully examined the authors' responses to the initial review and notes that the manuscript still lacks a critical discussion of alternative interpretations of the cosmogenic data. In addition, no figure or map has been provided to visually synthesize the paleoclimatic forces and geological processes proposed to have contributed to aridity during the Miocene–Oligocene.

For these reasons, two essential revisions remain necessary before the manuscript can be considered for publication:

1. Clarification of the study's hypothetical nature.

In their rebuttal, the authors reiterate the arguments of the original manuscript while acknowledging the hypothetical character of the proposed model. Although the reasoning presented is plausible, it cannot be regarded as definitive. It is therefore essential that the tentative nature of the interpretation be explicitly conveyed in the title and abstract, rather than being confined to the discussion.

2. Clear framing as a reassessment of previous work.

Although the authors demonstrate that their dataset substantially expands upon that of Dunai et al. (2005), the manuscript would benefit from a clearer statement in the introduction that the study constitutes a revisiting of the site. This contextualization would significantly aid readers.

Version 3:

Reviewer comments:

Reviewer #5

(Remarks to the Author)

Dear authors,

The main corrections requested by this reviewer have been addressed in the latest version prepared by the authors. I now propose that the article be published in Nature Communications.

Sincerely,

Reviewers' comments:

We sincerely thank the reviewers for their thoughtful and constructive comments. Below, we provide a detailed, line-by-line response to each point raised. For clarity:

- Reviewer comments are shown in italics.
- Our responses follow immediately after each comment in grey colour.
- All corresponding changes have been incorporated into the revised manuscript, where modifications are marked for easy reference.

Reviewer #1 (Remarks to the Author):

*Review of Ritter-Prinz et al., "Eocene aridification of the Atacama Deserts' hyperarid core."
Prepared by Marissa Tremblay, 2025.07.07*

This manuscript by Ritter-Prinz and coauthors presents new cosmogenic ^{21}Ne measurements from quartz clasts exposed in the low-relief alluvial surfaces the Chilean Coastal Cordillera as well as new U-Pb ages of tephra deposits located beneath these surfaces. Combined with previous cosmogenic ^{21}Ne measurements and other observations from the area, the authors argue that the onset of hyperarid conditions in this part of the Atacama Desert began in the Eocene, which is significantly earlier than the previously suggested Miocene onset. The authors hypothesize that Eocene hyperaridity could be linked to cooling after the Early Eocene Climatic Optimum and a stronger proto-Humboldt Current.

Overall, I found this paper to be well written, and the presentation of integrated exposure ^{21}Ne ages as old as the Eocene is an exciting result. I believe this is a valuable contribution that challenges our current understanding of the onset of hyperarid conditions in the Atacama, and it will inspire additional research on what the mechanisms for such an early onset of hyperaridity might be. My comments on the manuscript are relatively minor and should be straightforward to address. I recommend publication following minor revisions.

General comments:

Comment 1.1: Uplift scenarios: The uplift scenarios tested, as shown in Datatable_05_Uplift Models, all have the sample sites reaching their current elevations yesterday. An uplift scenario like this will tend to yield older exposure ages, simply because the samples will spend most of their exposure histories at lower elevations where the production rates are lower. I am not suggesting that these uplift models are incorrect, but it would be helpful if the authors could discuss why uplift of the Coastal Cordillera could not have occurred earlier, such that the surfaces they studied sat near present-day elevations for some prolonged period of time (~Myr). For example, citing geodetic observations or observations from coastal terrace uplift to support the argument that the Coastal Cordillera is experiencing present-day/recent uplift would be helpful.

Response 1.1: Thank you for pointing out the uncertainty in the uplift history of the Coastal Cordillera. We added additional references on current uplift rate estimates in the supplementary

Information file, indicating that uplift of the Coastal Cordillera is still an ongoing process. However, we want to point out that these are very recent observations and data. Indeed, the exposure age/durations will change towards younger ages/ lower durations if the sample site is at higher elevations. Based on the recent published knowledge and in the absence of any other uplift information (as stated in the method section and supplementary information), we apply the proposed 40m/Ma uplift rate for the Coastal Cordillera. We stated in the supplementary/method section, that a faster or slower uplift rate would change the exposure ages/duration by a maximum of 7.3% towards older or up to 15.4% towards younger ages. Applying these ranges, our dataset still provides old exposure ages / long exposure durations of clasts on the Earth's surface.

Comment 1.2: Data presentation: I think Figure 1 would be much stronger if there were a panel with the individual clast cosmogenic ^{21}Ne ages included, rather than only showing the individual ages in Extended Data Fig. 6A. If the figure is limited in space, I recommend removing the oxygen isotope curve and simply including the climatic events referred to in the text (EECO, Oi-1 glaciation) with the bar charts for the ice sheet chronology that appears beneath the O isotope curve.

Response 1.2: We agree. We moved the detailed Figure with the individual ^{21}Ne ages into the manuscript.

Comment 1.3: Does the relative probability plot in figure 2A include both the ^{21}Ne ages from the depositional surface and from the catchment to the south (i.e., the separate relative probability plots in 6A and 6B)? If so it might be helpful to state that in the caption for Extended Data Fig. 6, as I was only comparing Figure 1 and Extended Data Fig. 6A originally and was confused about where the subtle differences between the two were coming from.

Response 1.3: We agree. With the implementation of the detailed ^{21}Ne age figure (now Fig. 2 in the manuscript) into the manuscript and modification of the figure caption of Fig. 3, we solved this issue. Line 399: "(A) Relative probability plot of **all** new and previously published modelled cosmogenic Ne ages of clasts from the Jazpampa surfaces."

Comment 1.4: The authors should include an extended data figure that presents their new tephra U-Pb data (probably the compilation of U-Pb age PDFs, with the MDA annotated), rather than having these data only appear in the supplementary Excel spreadsheet.

Response 1.4: We agree and added this plot to the extended dataset figures. Now Extended Data Fig. 7 shows the probability density plot of ^{206}Pb ^{238}U data of both samples TIL22-02 and TIL22-03.

Comment 1.5: Finally, I think the authors should add the nucleogenic-air mixing line to Extended Data Fig. 7.

Response 1.5: We added the pathways of crustal and MORB (mantle Ne) to the triple isotope plot, as well as nucleogenic reactions producing those trends. Trends are taken from Kennedy et al. 1990 and Sarda et al. 1998, compiled in Niedermann 2002. We modified the Extended Data Fig. Caption: ... Trends of crustal and MORB Ne and reactions are taken from⁷³. 73 → Niedermann 2002

Line-by-line comments:

Line 22: Later in the manuscript on line 59, the authors state they have measured ^{21}Ne in only 122 quartz clasts. Which is correct?

In the abstract we state the total amount of ^{21}Ne ages/concentrations (135) used to establish this record, this also includes data from Dunai et al. 2005. We measured new unpublished ^{21}Ne concentrations of 122 clasts. In total 135 ^{21}Ne concentrations of clasts are used for the compilation.

Line 50: The word 'the' is needed before 'Coastal Cordillera'.

Added.

Line 174: I don't think it's a fair statement to say there is a near absence of clasts with exposure ages less than 10 Ma. From Datatable_04_Ne_Ages in the Excel spreadsheet provided, I count 34 clasts with exposure ages less than or equal to 10 Ma. For comparison, there are only 9 clasts with exposure ages in the data table that are greater than or equal to 34 Ma (i.e., Eocene or older).

We agree and modified this sentence to Line 193: "The low number of clasts with exposure ages of less than 10 Ma...."

Line 324: Is this fault scarp shown in the satellite imagery (I'm fairly certain it is)? If so, I recommended annotating the imagery with its location, like how you've noted the location of the Quedabra de Tiliviche.

We agree. We added all important fault scarps, and if existing also published names, see modified Fig. 1.

Reviewer #2 (Remarks to the Author):

Comment 2.1: This is an interesting dataset that is worthy of publication but may be in a longer format paper where a more detailed discussion of the dataset can be presented and the interpretations outlined more clearly.

Response 2.1: We now more explain and answer the question raised by the reviewer 2 in the manuscript and make use of the total length of Nature Communication article (first submitted manuscript ~2500 words)

Comment 2.2: Introduction should be written with a clear narrative – at the moment it jumps around from present day to rock record, and general to detailed from paragraph to paragraph. The information is there but needs to follow more coherently.

Response 2.2: We appreciate the reviewer's observation. The structure of the Introduction was intentionally designed to move from broad, present-day observations to more detailed interpretations of the rock record. Nevertheless, we have reviewed the flow to ensure that the transitions between these sections are clearer.

In paragraphs (Line 30-62), we summarize recent research on the evolution of aridity in the Atacama Desert and the spatial context of these studies. This is followed by our main research question, which builds on single cosmogenic nuclide exposure ages indicating earlier onsets of hyperaridity. We relate these isolated findings to additional lines of evidence and discuss the possible forcing factors that could have driven the early aridification of the modern hyperarid core of the Atacama Desert.

A short paragraph (Lines 63–70) then outlines the scope of this study, including the study material, methods, and key outcomes. The final paragraph (Lines 71–83) introduces the study area, first from a geological–tectonic perspective and then from a geomorphic standpoint, to provide the necessary framework and engage the reader.

In response to the reviewer's comment, we have also shortened this final paragraph and moved it toward the beginning of the section (now Lines 34–38), as the discussion of present-day precipitation had already been addressed earlier.

Comment 2.3: L39 remove 'Cenozoic' as record is back to 150 Ma

Response 2.3: We changed it accordingly.

Comment 2.4: Some clarity regarding the exact geomorphic relationships would be good. The sampled quartz clasts occur on and above dated tephra and the Azapa Formation of late Oligocene-Miocene age. Could the authors clarify how the exposure ages of the quartz clasts are older than the sediments on which they occur?

Response 2.4: We acknowledge the reviewer's comment. In the revised manuscript, we have clarified the geomorphic position of the sampled surfaces, the origin of the Azapa Formation sediments, and the stratigraphic and spatial relationships between the different formations, tephra layers, and sampled surfaces.

Line 92-99: "The ashes cap the Azapa Formation (rounded fluvial gravels, Extended Data Fig. 3,4), a regionally thick sediment sequence that was deposited around the Oligocene-Miocene transition^{15,33-35}(Fig. 1 A, B). The thick Azapa Formation strata are linked to the synchronous deposits of the Lower Moquegue Formation, the lower part of the Altos de Pica Formation, the Tambores Formation; and the Calama Formation^{17,40}. These deposits are the erosion products from the uplifting Andes to the east⁴¹ and were predominantly deposited within the Central Depression, where they partially infilled

basins and depressions that drained to the Pacific Ocean^{15,42}. These sediments record a non-local (allochthonous) climate signal.”

Specifically, we now describe in more detail how the sampled quartz clasts rest on erosional surfaces developed on and above the Azapa Formation and overlying dated tephra. We emphasize that these clasts represent material derived from local catchments within the Coastal Cordillera.

Line 101-106: “The Azapa and tephra deposits are overlain by low-relief surfaces that have been effectively protected from runoff from the Precordillera and the emerging Andes to the east since deposition (Fig. 1 A, B). Consequently, the quartz clasts sampled from site CH04-5 (Fig. 1B) must have originated from catchments within the Coastal Cordillera¹⁵, likely from veins in bedrock exposed on a local topographic high to the south (Fig. 1 B, C, yellow dashed line). “

We also clarify that although the ‘modelled’ exposure ages of the quartz clasts are older than the depositional ages of the Azapa sediments and tephra, this can be explained by inheritance of cosmogenic nuclides acquired during exhumation and transport prior to final deposition.

Line 145-149: “The younger age constrains the earliest time that clasts could have been deposited on the overlying low-relief surface. The majority of clasts, however, yield modelled exposure ages that are significantly older than the formation of the surface. Consequently, the clasts must have acquired cosmogenic Ne prior to final deposition during slow exhumation and transport. “

We acknowledge the reviewer’s concern regarding the apparent inconsistency between the exposure ages of the quartz clasts and the ages of the underlying Azapa Formation and tephra. In light of this and similar comments from other reviewers (see reviewer 3), we have revised the terminology used in the manuscript to improve clarity. Specifically, we now refer to the measured cosmogenic nuclide signal as exposure duration rather than exposure age.

This change better reflects the nature of the cosmogenic ²¹Ne concentrations measured in the quartz clasts, which integrate the entire history of exposure to cosmic rays—including periods of surface exposure, exhumation, limited transport, and final deposition. The term exposure duration thus more accurately captures the cumulative production of ²¹Ne in the quartz clasts and avoids implying that the measured values represent a single, uninterrupted period of surface exposure.

We believe this revised terminology helps to clarify the geomorphic interpretation and resolves the confusion regarding inherited nuclide components in relation to the depositional ages of the Azapa Formation and tephra.

Line 149-158: “Thus, to avoid confusion between clast deposition age and the time the clast has been at the Earth surface, we use the term exposure duration when referring to the measured concentrations of cosmogenic ²¹Ne in quartz clasts. This more accurately represents the total time a clast has been exposed to cosmic rays, and includes exhumation, transport, shallow burial and final deposition. This approach acknowledges that the cosmogenic ²¹Ne concentrations reflect the integrated production history rather than a single, continuous episode of exposure at their present position. Thus, the very long exposure durations recorded by the clasts from the Jazpampa surfaces imply that the region preserves a record of extremely slow landscape development consistent with the onset of intense aridification significantly earlier than the Early to Mid-Miocene^{15,16,19,27}. “

Comment 2.5: If, as noted, the area is hyper-arid it is unlikely that the quartz clasts have been transported any significant distance so how did they end up on the Azapa/tephra? A model illustrating how the quartz clasts arrived at their final resting place would be useful to explain to the reader. Is the argument that they have been exposed in the Coastal Cordillera since the Eocene and were then periodically reworked onto the Azapa/tephra surface but retained their exposure age?

Transport distance of 5 km or more are suggested in Fig. 1. Some comment on how much reworking/burial would need to take place to change the exposure age would be useful.

Response 2.5: As clarified above and in the revised manuscript (see definition of exposure duration), the measured ^{21}Ne concentrations in quartz clasts reflect the integrated history of cosmic-ray exposure rather than a single period of surface stability. Because ^{21}Ne is a stable cosmogenic nuclide and does not decay (unlike ^{10}Be), temporary burial or limited reworking will not reduce the accumulated ^{21}Ne concentration once produced. Therefore, even if the clasts were periodically reworked or buried beneath sediment layers, their inherited ^{21}Ne signal would be preserved.

The exposure durations we report thus represent the cumulative exposure history of the clasts, consistent with a model in which material originally exhumed and exposed in the Coastal Cordillera was episodically reworked/transported and deposited onto the Azapa Formation and tephra surfaces, retaining their previously acquired cosmogenic inventory.

As mentioned in the manuscript.

Line 106-108: "In this scenario, the transport of the clasts relied exclusively on local precipitation¹⁵. Episodic sediment transport in response to localised torrential rain events, which are well documented in the Atacama region^{45,46}."

Intermittent phases of sufficient precipitation allow surface activity, i.e. the transport of clasts from the catchment area to the final deposition. This does not imply direct transport, but clasts will be moved subsequently from time to time until they are at the final deposition site on the Jazpampa surfaces.

Comment 2.6: Sample PI-17 is not in the catchment to the main sample set and it's not easy to get a drainage system to link to the sample set from PI-06 either. Given that there has been little or no erosion since the Eocene drainage systems should be broadly similar (with the exception of the NE-SW faults which are younger), such that it is not easy to clearly link the samples in the catchments to those on the tephra/Azapa surface.

Response 2.6: We apologize for the potential misunderstanding and wrong wording. We included this sampling site, as it also represents the source area of the high concentration clasts. We do not intend to link those ^{21}Ne concentrations of clasts to the depositional surface we discuss in the manuscript. This sample sites serves as another example for the high concentration source areas within the Coastal Cordillera. We rephrase the introduction part of those two samples sites, clearly indicating which one represents the headwater of the supplying catchment for depositional Jazpampa surfaces further to the north.

Line 115-121: "Site PI-06 is the headwater of the catchment that has fed clasts to the Jazpampa surfaces (Fig. 1). Surface PI17-04 is a similar area from the same isolated topographic high. Both sites are locally the highest topographic points, typically ~300 m higher than the Jazpampa surfaces, and are rimmed by outcrops of Jurassic volcanic rocks⁴⁸ that are the source of the clasts. Both sample sites provide information about the exposure of clasts from the isolated source region within the Coastal Cordillera, that are not affected by any sediment transport sourced from the Andes to the east."

Comment 2.7: The conglomerates of the Azapa Formation clearly indicate that it was wetter at times in the study area in the Oligo- Miocene. How do the authors reconcile the deposition of the Azapa Formation in the Miocene with hyperaridity from the Eocene onwards.

Response 2.7: We refer to the response 2.4, where we describe more in detail the Azapa Formation and its source area.

Comment 2.8: The fact that supergene ages also extend back into the Eocene (Fig. 2) indicates that a hyper-arid climate was in place in areas of the Atacama since that time period, so the fact that there are clasts with an Eocene age of exposure is not unexpected, but the fact that they have been preserved in some areas is definitely of interest.

Response 2.8: Supergene copper mineralization provides important constraints on the timing of the transition from arid to hyper-arid conditions in the Atacama Desert, as their formation typically requires mean annual precipitation (MAP) above approximately 100-120 mm/yr (Arancibia et al. 2006, Alpers and Brimhall, 1988; Whitlow, 1990; Sillitoe and McKee, 1996, Evenstar et al. 2024).

Based on published data (Arancibia et al. 2006, Alpers and Brimhall, 1988; Whitlow, 1990; Sillitoe and McKee, 1996, Evenstar et al. 2024), the majority of supergene copper mineralization ages in northern Chile are of Early to mid-Miocene age and are therefore interpreted to reflect the onset of hyper-arid conditions, with MAP well below ~100 mm/yr. Supergene mineralization ages of Eocene age are comparatively rare and, in the context of the numerous Early to mid-Miocene ages, are not generally interpreted as indicating the regional transition to hyper-aridity. Rather, they represent localized conditions in parts of the Precordillera to the east of our study area.

In contrast, cosmogenic nuclide concentrations provide a complementary perspective, recording the persistence (or absence) of fluvial surface activity and surface exposure under progressively drier conditions.

Comment 2.9: A possible origin for onset of Eocene hyperaridity could be related to eastward expansion of the Central Andes. Uplift in the Subandean Zone/Eastern Cordillera is the principal orographic control in the Central Andes and was most likely initiated around 50 Ma (see Evenstar et al 2023 EPSL for more details).

Response 2.9: We thank the reviewer for this constructive comment. According to Evenstar et al. (2023, Earth and Planetary Science Letters), uplift in the Central Andes was diachronous and propagated eastward through time. Their synthesis indicates that initial surface uplift in the Western Cordillera began in the Oligocene, with approximately 50% of present-day elevations achieved prior to the Mid-Miocene. Uplift of the Altiplano followed, initiating in the Early Miocene, while the Eastern Cordillera/Subandean Zone experienced significant uplift from the Mid-Late Miocene onwards.

The evolution of aridity closely tracks this eastward migration of surface uplift, although lacking behind. Evenstar et al. (2023) show that supergene copper mineralisation—which requires mean annual precipitation (MAP) of 100–200 mm/yr—marks periods of semi-arid to arid conditions, whereas the cessation of supergene enrichment indicates a decline in MAP below ~100 mm/yr, reflecting a shift to hyperaridity. This transition to hyperarid conditions was diachronous, occurring by the Mid-Miocene in the Atacama Desert (largely only in the Precordillera) and substantially later, during the Late Miocene, along the western flank of the Western Cordillera.

Therefore, the aridification driven by the ‘orogenic–orographic feedback effect’ described by Evenstar et al. (2023) postdates the onset of hyperarid conditions inferred from cosmogenic nuclide data in the Coastal Cordillera. This suggests that, while Andean uplift influenced the progressive eastward expansion of aridity, the establishment of arid to hyperarid conditions in the Coastal Cordillera likely occurred earlier and was more closely linked to global climate cooling following the Early Eocene Climate Optimum. In summary, the orographic–orogenic feedback associated with Andean uplift likely intensified and maintained hyperaridity inland, but the initial onset of hyperaridity in the Coastal Cordillera preceded major Andean surface uplift.

We have revised the manuscript to reflect these insights and to acknowledge the potential orographic effects on the evolution of aridity across the Andean margin.

Line 202-207: "Aridification driven by the 'orogenic–orographic feedback effect' described by Evenstar, et al.²⁴ post-dates the onset of hyperarid conditions inferred from long-lived clasts in the Coastal Cordillera. This suggests that, while Andean uplift may have driven the progressive eastward expansion of aridity into the Precordillera, Western Cordillera and Altiplano Plateau, the establishment of predominantly hyperarid conditions in the Coastal Cordillera likely occurred significantly earlier."

In conclusion this is a very interesting manuscript which raises some fascinating questions about the history of hyper-aridity in the Atacama Desert. However, without further documentation and explanation it is not clear how the clasts have retained their exposure ages whilst being transposed onto a Miocene surface and needs to be more clearly explained/documentated. The transport of clasts significant distances without exposure ages being rest needs to be explained. The fact that hyperaridity in the core of the Atacama is already known to extend back to the Eocene based on supergene enrichment ages indicates that the implications are not as great as suggested.

We thank the reviewer for their detailed review and issues raised. With the above-mentioned explanations and change in the revised manuscript, we believe that the additional clarifications, and revisions we have made fully address the concerns raised and significantly strengthen the manuscript. We hope that these improvements demonstrate both the robustness of our conclusions and the broad significance of our findings.

Reviewer #3 (Remarks to the Author):

This is a well-written paper presenting new exposure ages to support an early hyper-aridification of the Atacama Desert. The introduction clearly explains the global importance of this topic. This contribution builds on the Dunai et al. (2005) Geology manuscript and extends those findings to suggest an earlier onset of hyper-aridity. Like the Dunai et al. paper, this paper uses cosmogenic ^{21}Ne to model surface exposure ages of clasts and uses these dates to constrain the timing of hyper-aridity.

Comment 3.1: And, like the Dunai et al. paper, this one suffers from the uncertainty of the role of ^{21}Ne inheritance in the measured concentrations of the clasts. Namely, that it's impossible to know how much of the ^{21}Ne was produced in the clasts in the sampled surface exposure versus how much was produced in the rock at some point in its near-surface exposure history. This paper addresses this uncertainty directly but can offer only suggestive explanations for why their modelled ages are in fact surface exposure ages without any inheritance.

Response 3.1:

We appreciate the reviewer's insightful and critical assessment regarding the fundamental challenge of ^{21}Ne inheritance in our clast concentration measurements. First of all, we have to point out that the reviewer accidentally did not recognize our statements in the manuscript about the relation of the measured ^{21}Ne concentration and exposure and inheritance prior to final deposition. We stated:

Line 131-133: "Such high concentrations integrate the long-term exposure of the quartz clasts to cosmic rays at the surface ($< 3 \text{ m}^1$), including initial exhumation from the source bedrock and subsequent transport to, and deposition at, the final site."

Due to this, in our original manuscript, we incorporated and sampled additional data to constrain any potential inheritance by including independent geochronological data of the sediments beneath the sampled sediment surfaces.

Line 143-145: "New U-Pb zircon dating of the lowermost and uppermost tephra layers yield ages of $25.17 \pm 0.48 \text{ Ma}$ ($n=4$) and $19.01 \pm 0.83 \text{ Ma}$ ($n=4$) respectively (Supplementary Datafile_2_Tephra_Data)."

Moreover, we state that our new U-Pb tephra ages give maximum final exposure ages for the quartz clasts:

Line 145-146: "The youngest age constraint provides the earliest time that clasts could have been deposited on the overlying low-relief surface."

In combination we critically evaluate our ^{21}Ne concentrations and their modelled exposure ages as being affected by pre-exposure prior to final deposition on the sampled sediment surface:

Line 146-148: "The majority of clasts, however, yield modelled exposure ages that are significantly older than the formation of the surface. Consequently, the clasts must have acquired cosmogenic Ne prior to final deposition during slow exhumation and transport."

*We never stated, "modelled ages are in fact surface exposure ages **without any** inheritance" (citation from reviewer statement at the top). In the contrary, as mentioned and cited above. We do not know, how the reviewer came to this wrong statement given the detailed interpretation and discussion of our data in the manuscript.*

We believe there may have been a terminological misunderstanding, although we made extra statements (see above, and Line 131-133). To decisively pinpoint and underline our full awareness of the inheritance problematic, we have performed the following revision in accordance with a comment raised by reviewer 2 (see above Comment 2.4): We have replaced the term "exposure age" with "exposure duration" after we discussed the inheritance problem throughout the remaining manuscript. This revised terminology clarifies that we are calculating a time span that these clasts were exposed to cosmic rays on Earth, which reflects the total integrated nuclide concentration, rather than a definitive depositional age. This aligns perfectly with our detailed discussion on inheritance and the constraints provided by our new independent tephra ages. This amendment, alongside the detailed existing discussion and supporting data, should effectively demonstrate that we are fully conscious of the limitations imposed by inherited nuclides and accurately communicates the scope of our findings.

Line 149-155: "Thus, to avoid confusion between clast deposition age and the time the clast has been at the Earth surface, we use the term exposure duration when referring to the measured concentrations of cosmogenic ^{21}Ne in quartz clasts. This more accurately represents the total time a clast has been exposed to cosmic rays, and includes exhumation, transport, shallow burial and final deposition. This approach acknowledges that the cosmogenic ^{21}Ne concentrations reflect the integrated production history rather than a single, continuous episode of exposure at their present position."

Comment 3.2: They also use an incomplete and carefully selected set of references, rather than a comprehensive set to support their suggestions. Their suggestive (and speculative) explanation, coupled with the correlation with supergene enrichment ages leads the authors to forward their results in a definitive, rather than speculative, way.

Response 3.2: We thank the reviewer for their critical engagement with our supporting literature and the subsequent interpretation of the exposure duration data. This feedback addresses a core challenge in regional geochronology: synthesizing sparse or disparate chronological markers.

Our aim was to construct an argument using the most complete set of published paleoclimatic, aridity, and geological evolution data available for the specific study region. We acknowledge that the current published geological and proxy record for this area is inherently limited, especially for time scales beyond the Miocene, which may create the perception of a selective bibliography.

We incorporated all existing published data for the time range to establish the regional context. Had additional studies or proxies been available, particularly those that directly address the specific time frames or geological constraints discussed, we would have certainly used them to enrich our discussion and test our interpretations. We would be grateful if the reviewer could specify any missing published studies or alternative proxies that he feels would significantly alter or strengthen our current synthesis. Without such guidance, we cannot fully address the general critique of the reference set.

Regarding the interpretation's tone—being seen as "definitive, rather than speculative"—we agree that clarity on the level of certainty is paramount. Our discussion was intended to present the most plausible and evidence-based hypothesis for regional geological evolution by integrating our newly derived ^{21}Ne exposure duration data with the few existing chronologies, such as the supergene enrichment ages.

We intended for this correlation to be seen as a hypothesis-driven integration of the only available constraints, which, while suggestive, naturally carries an inherent degree of uncertainty due to the data scarcity mentioned above.

We have reviewed and refined the language within the discussion to ensure that the integrated interpretation is clearly framed as a working hypothesis supported by the current totality of evidence, rather than an unassailable conclusion. We remain fully open to correcting our interpretation should new, high-quality chronological data become available, or if the reviewer can be more explicit in which peer-reviewed publications we missed and should include in our manuscript.

Comment 3.3: There are simply too many of the same flaws that were articulated in rebuttals of the Dunai et al. (2005) findings in this paper.

Response 3.3: We must respectfully state that we cannot fully address this critique without specific details. To make a scientifically valid revision based on this feedback, we kindly request the reviewer to identify and cite one or two specific publications that articulate the "flaws" they perceive to be replicated in our work.

We cannot currently follow the assertion that our paper suffers from the same issues, as our methodology was explicitly designed to mitigate known challenges:

- 1. Acknowledging Inheritance: As noted in our previous response, we explicitly discuss the accumulation of ^{21}Ne as a cumulative exposure duration rather than an absolute depositional age, directly addressing the core inheritance challenge.*
- 2. Mitigation via Independent Data: The key strength of our approach is the use of additional independent chronological constraints (the new tephra ages) which allow us to test the plausibility of the ^{21}Ne durations.*

In fact, numerous subsequent studies have reinforced the utility of ^{21}Ne as a valuable geochronometer, particularly for dating old landscapes and extremely slow-eroding desert environments, provided that the necessary precautions—which we have applied—are followed.

We are eager to engage with any published, peer-reviewed rebuttal that challenges the specific methodological precautions we have taken but require clear citations to do so effectively.

We believe that all our considerations, along with the methodological care described in the revised manuscript, directly address any potential concerns and reinforce the validity of our conclusions.

Comment 3.4: The details of these explanations, as well as untangling the methodology used lend themselves better to a specialty journal than a high profile, broad audience journal such as Nature. These results deserve to be published and put into the scientific discourse such that they can be more fully debated and assessed. For this to happen, the details of the sample collection and analyses would need to be fully fleshed out in a longer and more detailed paper.

Response 3.4: We submitted this work to Nature Communications precisely because we believe the scientific implication of our new exposure duration data—which provides critical chronological constraints on extremely old landscape evolution, arid-phase climate dynamics, and the long-term stability of this region—transcends the scope of a specialized geochronology journal.

While the technical foundation relies on established cosmogenic nuclide dating protocols, the integration of these data with independent tephra ages to redefine regional landscape stability offers novel, high-impact insight relevant to a broad readership in Earth and Environmental Sciences. We believe Nature Communications is the ideal venue for disseminating these significant results, facilitating the broader debate and assessment that the reviewer rightly calls for.

Completeness of Methodology: We are confident that our manuscript, as currently structured, meets the rigorous standards for detail and conciseness mandated by Nature Communications.

The main text provides a clear, high-level overview of the sample collection and analytical approach, while comprehensive details concerning the sampling strategy, laboratory preparation, mass spectrometry measurements, and all calculation parameters are provided in the Supplementary Information. This dual structure is the standard mechanism by which high-profile journals balance depth and breadth.

It is noted that this comment regarding insufficient detail was not raised in prior critiques. The current assertion that the "details... would need to be fully fleshed out in a longer and more detailed paper" appears to be in direct contradiction with the journal's requirement for a focused main narrative.

For the purpose of improving the manuscript and facilitating the necessary scientific discourse, we must insist that the reviewer points to tangible deficiencies in the methods or data analysis—specifying which sections of the Supplementary Information or main text require expansion, correction, or clarification.

Unfortunately, the reviewer does not specify what is missing or unclear and his review stays vague and not precise. With this we cannot engage, correct and modify our manuscript.

Comment 3.5: Similarly, an articulation of a more thorough explanation of how these results complement and build on the Dunai et al. (2005) data would need to happen as this paper often reads like a repeat of that paper.

Response 3.5: We respectfully disagree with the implication that our study merely repeats the work of Dunai et al. (2005). As noted in the comments from Reviewer 1 and Reviewer 2, the idea, goals, and overall outline of the study were found to be clear and understandable. In our manuscript, we explicitly explained the rationale for the additional work, why it was necessary for this new publication, and how it extends the previous study.

It is evident that our data not only complements Dunai et al. (2005) but also represents a clear advance in research, with important implications. Any apparent repetition from Dunai et al. (2005) arises from the fact that we revisited the original sampling site and performed additional analyses on both the original and new sites, using an expanded dataset. This approach was necessary to build upon the previous study and to validate and extend its conclusions.

Some overlap in background and description is therefore unavoidable, but it serves to contextualize the new findings. We respectfully suggest that this concern may not fully consider the detailed explanations and scope of the additional analyses presented, which clearly demonstrate the novelty and significance of this work.

While we welcome constructive critique, the nature of some comments implies that the reviewer may not have fully engaged with the manuscript or its analyses. In particular, the criticisms appear to target either the methodology employed or the research team, rather than the substantive content of the work. We have carefully addressed all technical and methodological points, and we believe the manuscript clearly demonstrates the validity and novelty of our work.

Unfortunately, I cannot recommend publishing the paper in Nature.

We have carefully addressed all technical and methodological points, and we believe the manuscript clearly demonstrates the validity and novelty of our work (see also comments from reviewer 1 and 2). We hope that with the response, answers and rebuttals, we can maybe convince the reviewer, that our measured ^{21}Ne concentrations and their interpretation are valid.

Reviewer #1 (Remarks to the Author):

*Review of Ritter-Prinz et al., "Eocene aridification of the Atacama Deserts' hyperarid core."
Prepared by Marissa Tremblay, 2026.01.23*

My comments on the original manuscript by Ritter-Prinz and coauthors were minor, and I find the authors have done a sufficient job answering my queries and critiques. I also think the authors have done a good job of responding to the comments of the other two reviewers and revising the manuscript accordingly. I also found reviewer #3's criticisms of the original manuscript to be vague and therefore difficult to address. I only have minor follow up comments regarding the revised version:

Lines 18, 23, 55: To be consistent in your revisions, I recommend changing the phrasing in these places from "exposure ages" to "exposure durations" as well.

Response: We agree, however, in Line 18 and 55 we refer to the published data and the use of exposure ages prior to our study and therefore think we should keep the use exposure age here. In line 24 we modified exposure age to exposure duration and added the following:

"...modelled exposure durations (time exposed to cosmic rays on the Earth's surface) ..."

Line 108: You may consider revising to "rare, localised torrential rain events."

Response: Changed accordingly.

Line 146-147: You may consider revising to "modelled apparent exposure ages."

Response: Changed accordingly.

Line 212: Given the acknowledged uncertainties regarding the uplift history, I recommend rephrasing from "probably under-estimate the true exposure history" to "may underestimate the true exposure history."

Response: Changed accordingly.

Reviewer #4 (Remarks to the Author):

Ritter-Prinz et al. present a new dataset of exposure ages derived from clasts on low-relief alluvial surfaces in the Coastal Cordillera. They argue that many clasts record mid- to late-Eocene exposure durations, with implications for how low landscape-evolution rates may be. The results also push the onset of aridity back to well before the classic early Miocene Humboldt Current/Andean uplift explanation. I read the authors' response to the reviewers, and I think they have either addressed or plausibly responded to the raised concerns. The idea is creative, and I therefore recommend publication after addressing the following comments.

I comment on this manuscript primarily from a geomorphology and landscape-evolution perspective. I do not have specific expertise in cosmogenic-nuclide exposure dating, so my comments focus on the geomorphic interpretation, assumptions, and the clarity of the landscape-evolution argument rather than the technical details of the nuclide methods.

Comment 2.1: 1) *The manuscript uses qualitative phrasing (e.g., “very slow landscape evolution,” “long exposure durations”) to link the exposure-age results to hyper-arid conditions, but it does not clearly quantify how slow the landscape evolution would need to be. It would strengthen the argument to translate these qualitative statements into quantitative bounds. For example: how much net erosion and/or deposition could have occurred while still preserving the inferred exposure durations and the observed surface/clast characteristics?*

Response 2.1:

We thank the reviewer for this thoughtful comment and agree that translating qualitative statements into quantitative bounds would, in principle, strengthen the manuscript. However, after further consideration, we believe that deriving meaningful quantitative erosion rates from the cosmogenic nuclide dataset presented here is not feasible on the timescales and under the geomorphic conditions targeted by this study, and that attempting to do so may be misleading.

In landscapes undergoing steady erosion, long-lived radionuclides such as ^{10}Be and ^{26}Al are commonly used to quantify erosion rates over timescales of $\sim 10^4$ – 10^6 years (Dunai, 2010; von Blanckenburg, 2005). In such cases, measured nuclide concentrations reflect a balance between production, decay, and surface lowering, allowing erosion rates to be calculated when steady-state conditions apply. These approaches typically provide erosion rates on the order of m/Ma for slowly eroding desert surfaces (Mohren et al., 2020; Nishiizumi et al., 2005; Ritter et al., 2023).

However, a key assumption of this method is that clasts erode in-situ from underlying bedrock, or that they have done so recently; a caveat in the case of fluvial samples collected for the purpose of averaged upstream erosion rates. This assumption is unlikely to be satisfied here as the clasts analysed in this study are interpreted as having significant periods of transport before deposition at their current location. These samples were not collected with the explicit aim of deriving erosion rates. Had this been the motivation a substantially different type of sampling would have been required.

The observation of ^{10}Be concentrations at or near saturation implies that no significant transport or burial has occurred over several million years, yet the ^{21}Ne results show much older ages that require these samples to have had a significantly long prior history (from which no ^{10}Be has survived because of its decay). Prolonged surface stability over multi-Myr timescales implies that post-erosional ^{21}Ne production is substantial and cannot be resolved from the ^{21}Ne inventory produced during erosion from bedrock. Consequently, any apparent ^{21}Ne -based erosion rate conflates erosion, transport history, and surface residence time and does not correspond to a physically meaningful erosion rate.

In summary, while qualitative statements such as “very slow landscape evolution” are grounded in the quantitatively high exposure durations documented in the manuscript, the cosmogenic nuclide data do not permit robust conversion of these observations into erosion rates. The combination of saturated ^{10}Be concentrations, and high ^{21}Ne inventories makes it fundamentally unwise—and potentially misleading—to use these data to calculate erosion rates. We therefore conclude that statements of slow landscape evolution are appropriately based on very high exposure durations rather than on inferred erosion-rate estimates. They are qualitative but necessarily so.

We added in the manuscript Line 142-146:

“Near-saturated to saturated ^{10}Be concentrations, together with high ^{21}Ne inventories imply negligible or extremely low net clast erosion over at least the past few million years. Saturation or secular equilibrium of ^{10}Be requires that samples have remained at the surface, or surface lowering has been minimal over timescales of the past millions of years (~4-5Myr, the characteristic time to reach secular equilibrium).”

We added in the manuscript Line 163-166:

““Under such conditions, using the measured cosmogenic nuclide data to calculate quantitative erosion rates is not appropriate, as steady-state assumptions are violated and nuclide inventories gained during erosion from bedrock cannot be separated from post-erosion production. “

Comment 2.2: *Is there a way to benchmark the implied rates against modern analogs in hyperarid to arid deserts (or even semi-arid settings) to show whether the required erosion/deposition rates are uniquely consistent with hyperaridity? In other words, could the inferred low long-term rates instead reflect a system dominated by a small number of high-magnitude events superimposed on otherwise low background geomorphic activity, rather than requiring persistently hyperarid conditions?*

If there is no way to such quantification, I encourage the authors to acknowledge this limitation as it could be arid, semiarid climate with a small number of events with high-magnitude can do same geomorphic work, low rates of landscape evolution.

Response 2.2:

We thank the reviewer for this comment. Climatic classifications such as hyperarid, arid, or semi-arid are commonly used and defined by mean conditions averaged over longer timescales. Thus, even in modern hyperarid deserts (e.g., the Atacama Desert or Namib Desert), rare precipitation or erosional event can occur without altering the overall hyperarid climate state when considered over 10^3 – 10^6 -year timescales. Consequently, the extremely low long-term surface activity based on our ^{21}Ne data is consistent with a predominant hyperarid climate averaged over geological timescales, even if interrupted by infrequent precipitation events or phases.

As discussed in the manuscript, we interpret the landscape as having experienced predominantly hyperarid conditions over long timescales, interrupted by intermittent pluvial events capable of mobilizing sediment and clasts. Such events may occur on annual to centennial recurrence intervals, but when averaged over millennial to million-year timescales, their geomorphic effect remains still very small (otherwise high ^{21}Ne concentration cannot build up). We therefore acknowledge that cosmogenic-nuclide exposure durations and the implications this has for extremely low surface activity cannot distinguish uniquely between persistently hyperarid conditions (this would produce a narrow exposure duration distribution since the onset of persistent hyperarid conditions) and a hyperarid regime interrupted by rare, high-magnitude events superimposed on long periods of geomorphic quiescence.

We wrote in the manuscript and modified where not previously mentioned in detail, that a predominant hyperarid background climate existed which was frequently interrupted by ‘wetter’ phases capable of causing some, but very minor surface activity.

Comment 2.3: 2) *Null hypothesis: I think the authors should evaluate whether non-climatic processes such as clast recycling, intermittent burial and exhumation, and surface armoring could also explain the observed exposure ages.*

Response 2.3: We thank the reviewer for raising this important point. Regarding intermittent burial, exhumation, and clast recycling: we measured cosmogenic stable ^{21}Ne in quartz clasts. Because ^{21}Ne is stable and does not decay, its concentration integrates total residence in the uppermost few metres. In contrast to radionuclides such as ^{10}Be or ^{26}Al , intermittent burial or shielding does not alter previously accumulated ^{21}Ne concentrations. Consequently, processes such as temporary burial, exhumation, or clast recycling would not reduce the apparent ^{21}Ne -derived exposure durations in the way they would affect decaying cosmogenic nuclides.

In fact, burial of clasts, i.e. shielding from cosmic rays, would require even older exposure durations and longer transport durations. That is, the entire history—from bedrock erosion, through clast transport with intermittent and sufficient burial, to re-transport and final deposition—must have occurred over an even longer timescale. That means that our calculated exposure durations are minimum durations.

With respect to surface armouring, several studies from the Atacama Desert have documented the widespread development of gypsum surface covers derived from atmospheric dry deposition (e.g. Rech et al., 2019; Rech et al., 2003; Wang et al., 2015), followed by secondary modification (Hartley and May, 1998; May et al., 2020; Pfeiffer et al., 2021) under limited moisture availability (e.g., rare precipitation events, fog, or dew). This gypsum cover has been shown to play a key role in surface stabilization by armouring the landscape, reducing overland flow, and protecting surfaces from fluvial reworking during rare rainfall events (Jordan et al., 2015; May et al., 2020; Ritter et al., 2022). In addition, “born-at-the-surface” models (e.g. Wells et al., 1995) have been invoked to explain the persistence of clasts forming desert pavements while gypsum soils continue to accumulate beneath them (e.g. Fuchs et al., 2025; Wells et al., 1995).

These processes are thought to promote long-term surface preservation through cementation and crust formation in the upper decimetres of the gypsum soil, potentially leading to positive feedback in hyperarid environments (Ritter et al., 2022). Such feedback enhances the preservation of atmospheric deposits and contributes to so-called soil inflation (e.g. Fuchs et al., 2025; Wells et al., 1995). Once a critical (yet still poorly constrained) threshold is reached, an active topic of current research, the combined effect of gypsum accumulation and surface armouring can strongly inhibit erosion, resulting in exceptionally stable land surfaces (e.g. Ritter et al., 2022).

Overall, while surface armouring likely contributes to surface stability, it does not compromise the validity of the ^{21}Ne exposure ages. Instead, it is consistent with the long-term preservation of clasts and the absence of intense fluvial reworking/activity required to record the observed cosmogenic ^{21}Ne concentrations.

We added some information in the manuscript to meet the reviewer’s comment:

Line 206-219:

“The prolonged and predominantly hyperarid climate of the Atacama Desert promotes a positive feedback mechanism by enabling the accumulation and long-term preservation of atmospherically deposited gypsum dust, a process referred to as soil inflation and ubiquitous across gypsisols of the region (Ericksen, 1981; Ewing et al., 2006; Rech et al., 2003; Wang et al.,

2015). Widespread gypsum surface covers derived from atmospheric dry deposition (Wang et al., 2015) have been documented and are subsequently modified under limited moisture availability, such as rare precipitation events, fog, or dew (Hartley and May, 1998; May et al., 2020; Pfeiffer et al., 2021). These surfaces undergo induration through encrustation and cementation, which enhances sediment strength, armours the landscape, reduces overland flow, and protects surfaces from fluvial reworking during episodic rainfall (Ritter et al., 2022). “Born-at-the-surface” models further explain the persistence of pavement clasts while gypsum soils continue to accumulate beneath them (Fuchs et al., 2025; Wells et al., 1995). Together, these processes increase the critical precipitation threshold required to trigger surface activity and erosion (Ritter et al., 2022), thereby promoting long-term surface stability and supporting long clast exposure durations, consistent with cosmogenic ^{21}Ne concentrations and the interpretation of predominant hyperarid conditions over million-year timescales in parts of the Atacama Desert.”

Comment 2.4: 3) *Burial and exhumation: Could the authors conduct sensitivity tests to quantify how much intermittent burial (in terms of depth and duration) could occur while still reproducing the observed exposure age distribution?*

Response 2.4: Sensitivity tests of intermittent burial and exhumation are not applicable for the ^{21}Ne exposure durations presented here. The measured cosmogenic ^{21}Ne in quartz is a stable nuclide and therefore records the integrated duration of surface exposure irrespective of subsequent burial or exhumation, provided that burial depths are sufficient to fully shield the clasts from further cosmogenic production (i.e., on the order of ≥ 3 m of overburden).

Unlike radionuclides such as ^{10}Be or ^{26}Al , intermittent burial does not reduce or modify accumulated ^{21}Ne concentrations through radioactive decay. Consequently, variations in burial depth or duration would not alter the resulting ^{21}Ne -derived exposure age distribution, as long as production is effectively zero during burial.

We clarify this point above why intermittent burial and exhumation cannot bias the ^{21}Ne signal and why such sensitivity analyses are not applicable for stable-nuclide-based exposure ages/durations.

Overall, I think the authors have already addressed the previous reviewers' comments, and the manuscript is an important contribution to understanding the onset of aridity in one of the world's most important deserts. My comments are intended only to improve the geomorphic aspects of the paper, or at least to encourage the authors to acknowledge the study's limitations.

Best wishes,

Abdallah Zaki

We thank the reviewer for the additional geomorphic view on the manuscript and his comments.

Reviewer #5 (Remarks to the Author):

Review of Ritter-Prinz et al., "Eocene aridification of the Atacama Deserts' hyperarid core."

This review is based on the revised manuscript, taking into account the comments and refutations of three previous reviewers.

The article hypothesizes that the onset of aridity/hyperaridity in the Atacama Desert may have occurred much earlier (since the Eocene) than previously established (Oligocene-Miocene boundary). This chronological shift is based on the study of accumulated cosmogenic signals analyzed in clasts deposited on low-gradient, minimally eroded surfaces in one of the most arid sectors of the modern Atacama Desert. Most of the data are derived from a previously published study (Dunai et al., 2005). Once presenting their hypothesis, the authors propose that the primary driver of Eocene aridity was the global cooling that occurred prior to the Eocene Climate Optimum.

We have to correct the reviewer. The comment that '*most of the data are derived from...*' is not correct and does not reflect the immense new data acquisition and the progress of this study compared to the published work by Dunai et al. 2005. In Dunai et al. 2005, **12** single clasts ^{21}Ne concentration as well as one amalgamated sample were presented. In contrast, we added and extended the data set with more sample sites and quantitatively more sample data and now provide up to **122** new ^{21}Ne clast data among **additional ^{10}Be data**. Moreover, we added **zircon U-Pb LA ICP-MS** of the underlying tephra deposit.

Secondly, we have to correct the reviewer and their comment "*driver of Eocene aridity was the global cooling that occurred **prior** to the Eocene Climate Optimum*". We state correctly in the abstract line 27-28 (line numbers according to the revised manuscript):

"We postulate that global cooling **after** the Early Eocene Climatic Optimum was likely a key driver of regional aridification."

Line 233-234: "it is possible that global climate cooling **following** the Early Eocene Climate Optimum (EECO)5 was the trigger (Fig. 3A, D)."

Line 236: "a threshold **after** the EECO"

The available data are insufficient to present the hypothesis as an unequivocal thesis; it remains a weak proposal.

We would like to clarify that we do not present our interpretation as an unequivocal thesis. In the manuscript we state Line 222 "we hypothesize that the current dry core of the Atacama Desert has already experienced extreme aridity long before the early Miocene" derived from the available data and place them within the context of alternative explanations.

In particular, the manuscript includes a detailed discussion of potential mechanisms that could lead to high ^{21}Ne concentrations, including scenarios other than increasing aridity. Based on this critical evaluation and comparison of competing explanations, we conclude that an intensification of aridity, resulting in decreasing surface activity, is the most plausible explanation consistent with the analysed data obtained in this study. This conclusion is presented as a reasoned interpretation rather than a definitive claim.

We therefore respectfully disagree with the reviewer's comment that the hypothesis is presented as an '*unequivocal thesis*', and we encourage a careful rereading of the discussion section where uncertainties, limitations, and alternative scenarios are explicitly acknowledged.

From this reviewer's perspective, there are significant stratigraphic and sedimentological arguments that hinder the acceptance of the hypothesis and even render it somewhat contradictory:

Comment 3.1: 1. Stratigraphic Constraints: The deposition of the analyzed gravels on flat, uneroded surfaces certainly occurred after 12 and 25 Ma (Oligocene-Miocene boundary to Miocene), as evidenced by the age of the tephra underlying the gravels. This a priority and definitively constrains the age of both the deposit and the planation surface.

Response 3.1:

We thank the reviewer for this comment but note that it is based on a misunderstanding of both the tephra ages and the interpretation of the cosmogenic nuclide data. Unfortunately, we cannot follow the reviewer's statement that the deposition of clasts occurred after 12 (and 25) Ma. The presented tephra data, as mentioned in the manuscript indicate a deposition age of 19-25Ma.

Line 148-149:

"New U-Pb zircon ages of the lowermost and uppermost tephra layers are 25.17 ± 0.48 Ma (n=4) and 19.01 ± 0.83 Ma (n=4) respectively."

As stated repeatedly in the manuscript, we do not interpret the ^{21}Ne concentrations as dating the planation surface itself. The cosmogenic ^{21}Ne data record the integrated exposure history of individual quartz clasts, including exposure during bedrock erosion in the source area, transport, and residence prior to final deposition. Deposition of the gravels necessarily occurred after ~19 Ma, once the tephra was emplaced, and there is no contradiction between this stratigraphic constraint and the ^{21}Ne exposure durations.

The measured ^{21}Ne signal is therefore not controlled by the age of the flat surface but is dominantly inherited from prolonged exposure during slow erosion and transport in the southern source catchments, followed by final deposition and preservation on the surface above the dated tephra. Consequently, the stratigraphic ages provide a firm minimum age for gravel deposition, while the ^{21}Ne data constrain the cumulative exposure history of the clasts prior to and after deposition.

See for reference in the manuscript:

Line 147-148: "The quartz clasts from the Jazpampa surfaces were deposited after the eruption and deposition of the tephra exposed in Quebrada Tiliviche."

Line 150-151: "The younger age constrains the earliest time that clasts could have been deposited on the overlying low-relief surface."

Line 151-152: "The majority of clasts, however, yield modelled apparent exposure ages that are significantly older than the formation of the surface."

For the specific interpretation of exposure duration instead of exposure age, see line 154-163: "Thus, to avoid confusion between clast deposition age and the time the clast has been at the Earth surface, we use the term *exposure duration* when referring to the measured concentrations of cosmogenic ^{21}Ne in quartz clasts. This more accurately represents the total time a clast has been exposed to cosmic rays, and includes exhumation, transport, shallow burial and final deposition. This approach acknowledges that the cosmogenic ^{21}Ne concentrations reflect the integrated production history rather than a single, continuous episode of exposure at their present position."

Comment 3.2: 2. Transport Dynamics: Due to model requirements, the authors indicate that the spatial transport of clasts to the depositional area was short. It is stated that the materials deposited in the Miocene were transported within relatively closed and local sectors of the Coastal Range, isolated from inputs from the Precordillera and the Andes. This short-distance transport suggests it could not have been extended in time. Consequently, it is difficult to reconcile a ~25 Ma interval (the

duration between the Eocene and the lower Miocene boundary) to complete the transport from a hypothetical Eocene exhumation zone to the Miocene depositional site.

Response 3.2:

We thank the reviewer for this comment but respectfully disagree with the implied inconsistency between short transport distances and long transport durations.

Our interpretation is based on two independent observations: (i) the geomorphic setting of the depositional surface, which indicates spatially restricted transport within local catchments of the Coastal Cordillera, and (ii) the cosmogenic ^{21}Ne exposure duration distributions, which are comparable between clasts sampled on the depositional surface (see Fig 2A, Site A,B,C from Dunai et al.2005, CH-04-5, PI18) and those sampled directly within the source catchments (see Fig 2B, PI-06,PI17-04).

Short transport distance does not imply short transport duration. In hyperarid landscapes characterized by minimal fluvial connectivity, transport can occur through very slow, discontinuous processes (e.g., creep, rare low-efficiency runoff, or localized reworking), allowing clasts to remain exposed to cosmic rays for millions of years while moving only limited distances. The comparable exposure duration distributions (see Fig. 2) between source and depositional areas are consistent with such a regime and argue against efficient, rapid transport that would reset or significantly alter the cosmogenic inventory. Additionally we refer to numerous cosmogenic nuclide-based erosion rate studies from the Atacama Desert (Mohren et al., 2020; Nishiizumi et al., 2005; Placzek et al., 2014; Ritter et al., 2023) and from Antarctica (Margerison et al., 2004; Schäfer et al., 1999; Summerfield et al., 1999), which all put emphasis on extremely low erosion rates derived from cosmogenic nuclide concentrations, i.e. transport dynamics on timescale of millions of years.

We note that the reviewer states that it is “difficult to reconcile” a ~25 Ma interval for transport from an Eocene exhumation zone to the Miocene depositional site, but no alternative mechanism or dataset is provided that would contradict this interpretation. In the absence of evidence for higher transport efficiency, more extensive sediment routing, or younger exposure signatures, the simplest explanation consistent with the data is long-term persistence of very low geomorphic activity.

We therefore maintain that our interpretation is internally consistent, supported by multiple datasets, and does not conflict with the inferred short spatial transport distances.

Comment 3.3: 3. *Lack of Eocene Surfaces: The authors have not found—nor do they mention—flat/uneroded surfaces containing Eocene-aged deposits with clasts exposed since that period. Such a finding would constitute unequivocal evidence of Eocene hyperaridity. In fact, the absence of these deposits and surfaces suggests that, if they did form, they were likely eroded during a humid, non-arid Eocene period.*

Response 3.3: We note that this comment rests on a conceptual assumption about how cosmogenic exposure durations relate to preserved geomorphic surfaces. Our interpretation does not require the existence of an intact, flat Eocene-aged depositional surface that have remained uneroded to the present. On the contrary, the cosmogenic ^{21}Ne data indicate that surface activity has occurred continuously, albeit at extremely low rates, since at least the Eocene (see exposure duration distribution in Fig. 2). The reviewer already described this process, where since the Eocene surfaces are being eroded by intermittent pluvial phases during the predominant hyperarid background climate. If surfaces had been completely inactive since the Eocene, we would expect a much narrower or even singular exposure-age signal rather than the observed distribution.

In our framework, clasts acquire cosmogenic ^{21}Ne progressively as they are produced by very slow exhumation and erosion of bedrock and subsequently transported over long timescales under conditions of extremely low geomorphic efficiency, i.e. lack of sufficient fluvial activity. The absence of preserved, undeformed Eocene surfaces therefore does not imply erosion during a humid Eocene interval, but rather reflects ongoing, very slow surface lowering and reworking that prevents long-term preservation of discrete Eocene geomorphic markers while still allowing clasts to accumulate multi-million-year exposure histories.

Additional support for the existence of clast of Eocene to Oligocene exposure age/duration is provided by cosmogenic ^{21}Ne data from within the proposed hyperarid core of the Atacama Desert (Ritter et al., 2018). Ritter et al. (2018) report high ^{21}Ne concentrations corresponding to apparent clast exposure ages (or exposure durations) of ~35, 27, and 25 Ma (South Rio Loa 21.5°S), while Ritter et al. (2022) document similarly high apparent exposure ages of ~36, 25, and 23 Ma (Huara 20°S). Together, these results corroborate, spatial independent being studied at two different sites within the Coastal Cordillera (21.5°S, 20°S, and this study 19.3°S), the existence of Eocene to Oligocene exposed clasts within the hyperarid core of the Atacama Desert.

We therefore emphasize that the lack of preserved Eocene depositional surfaces is not evidence against long-term aridity. Rather, the cosmogenic exposure data are most consistent with a landscape undergoing very low and/or partially temporal zero erosion and transport since the Eocene, allowing clasts to remain exposed to cosmic rays for tens of millions of years. Surface activity is controlled by rare intermittent pluvial phase in the albeit predominant hyperarid climate (see Line 108-109: “Episodic sediment transport in response to rare, localised torrential rain events”).

I consider that is the Editor’s decision whether to accept a proposal based on the data provided. In its current form, it constitutes solely a working hypothesis. To be definitively proven, the authors’ idea requires additional study sites in the region, including the discovery and analysis of Eocene deposits and surfaces. Should it be published, the authors should adopt a more tentative tone in both the title and the abstract. A more suitable title would be: “Indications of the transition suggesting arid conditions in the Atacama Desert during the Eocene.”

Comment 3.4: *Beyond these primary considerations, the authors could improve the explanation of the causes for the proposed Eocene hyperaridity. This could be achieved by adding synoptic maps of the region illustrating the regional factors driving aridity, with separate maps for the Eocene and Miocene stages. Furthermore, the manuscript would be enriched by a discussion on the emergence, existence, and variability of SE and N-NE summer monsoons as mechanisms for moisture/precipitation delivery to Atacama latitudes, as demonstrated Sáez et al. (2016) for the Holocene.*

Response 3.4:

We thank the reviewer for this thoughtful suggestion. While we agree that a detailed paleoclimate synthesis including synoptic maps and a process-based analysis of atmospheric circulation would be valuable, the requested additions are beyond the scope of the present manuscript and beyond the specific expertise of the author team. Such analyses would require dedicated paleoclimate modeling and regional atmospheric reconstructions and are better suited to future interdisciplinary work.

Previous studies demonstrate (e.g. Houston, 2006; Houston and Hartley, 2003) that SE and N-NE summer monsoon systems exert a strong control on moisture delivery to the margins of the Atacama Desert (which receive also today much more precipitation compared to the hyperarid core, where our study site is located), with pronounced effects during the late Neogene and

Quaternary (e.g. Sáez et al., 2016). These systems primarily influence the eastern, northeastern and southern desert margins and show significant temporal variability (e.g. Houston, 2006; Houston and Hartley, 2003; Sáez et al., 2016). However, most of the studies are temporally limited and rarely been investigated on the timescales of this study.

Our study area is located near the boundary between these two moisture paths, where the influence of both SE and N–NE summer monsoons strongly diminishes (winter summer boundary, in Houston et al. 2006). As such, our study site represents an endmember setting of minimal precipitation input from either moisture source. This geographic position makes it particularly sensitive to long-term background aridity, but it also limits our ability to resolve changes in specific precipitation mechanisms or their temporal evolution relate to the mentioned monsoon.

With our dataset we are not able to differentiate between both precipitation regimes, and thus a detailed discussion about the “emergence, existence, and variability of SE and N-NE summer monsoons” is not possible.

Additional References cited in the reply letter:

- Dunai, T. J., 2010, *Cosmogenic Nuclides: Principles, concepts and applications in the Earth surface sciences*, Cambridge University Press.
- Ericksen, G. E., 1981, *Geology and origin of the Chilean nitrate deposits*: USGS, 1188.
- Ewing, S. A., Sutter, B., Owen, J., Nishiizumi, K., Sharp, W., Cliff, S. S., Perry, K., Dietrich, W. E., McKay, C. P., and Amundson, R., 2006, A threshold in soil formation at Earth's arid-hyperarid transition: *Geochim. Cosmochim. Acta*, v. 70, p. 5293-5322.
- Fuchs, M., Dietze, M., Brenning, A., Sauer, D., Schepanski, K., and Wagner, D., 2025, Desert pavements: A hidden key to Earth surface processes: *Earth Surface Processes and Landforms*, v. 50, no. 15, p. e70213.
- Hartley, A. J., and May, G., 1998, Miocene gypcretes from the Calama Basin, Northern Chile: *Sedimentology*, v. 45, p. 351-364.
- Houston, J., 2006, Variability of precipitation in the Atacama Desert: its causes and hydrological impact: *International Journal of Climatology*, v. 26, no. 15, p. 2181-2198.
- Houston, J., and Hartley, A. J., 2003, The Central Andean west-slope rainshadow and its potential contribution to the origin of hyper-aridity in the Atacama desert: *International Journal of Climatology*, v. 23, p. 1453-1464.
- Jordan, T., Riquelme, R., González, G., Herrera, C., Godfrey, L., Colucci, S., Gironás-León, J., Gamboa, C., Urrutia, J., and Tapia, L., Hydrological and geomorphological consequences of the extreme precipitation event of 24–26 March 2015, Chile, *in Proceedings XIV Congreso Geológico Chileno (La Serena)2015*.
- Margerison, H. R., Phillips, F. M., Stuart, F. M., and Sugden, D. E., 2004, Cosmogenic ³He concentrations in ancient flood deposits from the Coombs Hills, northern Dry Valleys, east Antarctica: interpreting exposure ages and erosion rates: *Earth Planet. Sci. Lett.*, v. 230, p. 163-175.
- May, S. M., Meine, L., Hoffmeister, D., Brill, D., Medialdea, A., Wennrich, V., Gröbner, M., Schulte, P., Steininger, F., Deprez, M., de Kock, T., and Bubenzer, O., 2020, Origin and timing of past hillslope activity in the hyper-arid core of the Atacama Desert–The formation of fine sediment lobes along the Chuculay Fault System, Northern Chile: *Global and Planetary Change*, v. 184, p. 103057.
- Mohren, J., Binnie, S. A., Ritter, B., and Dunai, T. J., 2020, Development of a steep erosional gradient over a short distance in the hyperarid core of the Atacama Desert, northern Chile: *Global and Planetary Change*, v. 184, p. 103068.

- Nishiizumi, K., Caffee, M. W., Finkel, R. C., Brimhall, G., and Mote, G., 2005, Remnants of a fossil alluvial fan landscape of Miocene age in the Atacama desert of northern Chile using cosmogenic nuclide exposure age dating: *Earth Planet. Sci. Lett.*, v. 237, p. 499-507.
- Pfeiffer, M., Morgan, A., Heimsath, A., Jordan, T., Howard, A., and Amundson, R., 2021, Century scale rainfall in the absolute Atacama Desert: Landscape response and implications for past and future rainfall: *Quaternary Science Reviews*, v. 254, p. 106797.
- Placzek, C., Granger, D. E., Matmon, A., Quade, J., and Ryb, U., 2014, Geomorphic process rates in the central Atacama Desert, Chile: Insights from cosmogenic nuclides and implications for the onset of hyperaridity: *American Journal of Science*, v. 314, no. 10, p. 1462-1512.
- Rech, J. A., Currie, B. S., Jordan, T. E., Riquelme, R., Lehmann, S. B., Kirk-Lawlor, N. E., Li, S., and Gooley, J. T., 2019, Massive middle Miocene gypsic paleosols in the Atacama Desert and the formation of the Central Andean rain-shadow: *Earth and Planetary Science Letters*, v. 506, p. 184-194.
- Rech, J. A., Quade, J., and Hart, W. S., 2003, Isotopic evidence for the source of Ca and S in soil gypsum, anhydrite and calcite in the Atacama Desert, Chile: *Geochim. Cosmochim. Acta*, v. 67, p. 575-586.
- Ritter, B., Diederich-Leicher, J. L., Binnie, S. A., Stuart, F. M., Wennrich, V., Bolten, A., and Dunai, T. J., 2022, Impact of CaSO₄-rich soil on Miocene surface preservation and Quaternary sinuous to meandering channel forms in the hyperarid Atacama Desert: *Scientific Reports*, v. 12, no. 1, p. 1-9.
- Ritter, B., Mohren, J., Binnie, S. A., Wennrich, V., Dunkl, I., Albert, R., Gerdes, A., LoBue, S., and Dunai, T. J., 2023, Shaping the Huara Intrusive Complex in the hyperarid Atacama Desert—Erosional near-stasis contrasting high topographic gradients: *Journal of Geophysical Research: Earth Surface*, p. e2022JF006986.
- Ritter, B., Stuart, F. M., Binnie, S. A., Gerdes, A., Wennrich, V., and Dunai, T. J., 2018, Neogene fluvial landscape evolution in the hyperarid core of the Atacama Desert: *Scientific Reports*, v. 8, no. 1, p. 13952.
- Sáez, A., Godfrey, L. V., Herrera, C., Chong, G., and Pueyo, J. J., 2016, Timing of wet episodes in Atacama Desert over the last 15 ka. The Groundwater Discharge Deposits (GWD) from Domeyko Range at 25° S: *Quaternary Science Reviews*, v. 145, p. 82-93.
- Schäfer, J. M., Ivy-Ochs, S., Wieler, R., Leya, I., Baur, H., Denton, G. H., and Schlüchter, C., 1999, Cosmogenic noble gas studies in the oldest landscape on Earth: surface exposure ages of the Dry Valleys, Antarctica: *Earth Planet. Sci. Lett.*, v. 167, p. 215-226.
- Summerfield, M. A., Stuart, F. M., Cockburn, H. A. P., Sudgen, D. E., Denton, G. H., Dunai, T. J., and Marhant, D. R., 1999, Long-term rates of denudation in the Dry Valleys, Transantarctic Mountains, southern Victoria Land, Antarctica based on in-situ produced cosmogenic ²¹Ne: *Geomorphology*, v. 27, p. 113-129.
- von Blanckenburg, F., 2005, The control mechanisms of erosion and weathering at basin scale from cosmogenic nuclides in river sediment: *Earth and Planetary Science Letters*, v. 237, no. 3-4, p. 462-479.
- Wang, F., Michalski, G., Seo, J.-H., Granger, D. E., Lifton, N., and Caffee, M., 2015, Beryllium-10 concentrations in the hyper-arid soils in the Atacama Desert, Chile: Implications for arid soil formation rates and El Niño driven changes in Pliocene precipitation: *Geochimica et Cosmochimica Acta*, v. 160, p. 227-242.
- Wells, S. G., McFadden, L. D., Poths, J., and Olinger, C. T., 1995, Cosmogenic ³He surface exposure dating of stone pavements: *Geology*, v. 23, p. 613-616.

Reviewers' comments:

We sincerely thank the reviewers for their thoughtful and constructive comments. Below, we provide a detailed, line-by-line response to each point raised. For clarity:

- Reviewer comments are shown in italics.
- Our responses follow immediately after each comment in grey colour, with changes to the manuscript made in green.
- All corresponding changes have been incorporated into the revised manuscript, where modifications are marked for easy reference.

Reviewer #4 (Remarks to the Author):

I have read the revised manuscript and the authors' responses to the comments raised by me and other reviewers. I appreciate the efforts made to address the concerns, and I think the manuscript is ready for publication. Thank you.

We thank the reviewer for the additional evaluation of our manuscript and appreciate that the revisions have satisfactorily addressed their comments and suggestions.

Reviewer #5 (Remarks to the Author):

Comment 2.1: This reviewer has carefully examined the authors' responses to the initial review and notes that the manuscript still lacks a critical discussion of alternative interpretations of the cosmogenic data.

Response 2.1: We thank the reviewer for this comment. However, the request for a "critical discussion of alternative interpretations" remains somewhat unspecific. In the original and revised manuscript, we have carefully considered and explicitly addressed all plausible mechanisms known to us that could account for the observed exceptionally high cosmogenic ^{21}Ne concentrations.

Specifically, we evaluate alternative explanations such as:

(i) non-cosmogenic ^{21}Ne production (e.g., nucleogenic contributions), which we shortly stated in the manuscript (Line 175-176) and in the Extended Data Figure Caption Fig. 7 (Line 483-486). In order to make it more visible, we added a sentence in the manuscript:

We modified and added Line 176-179:

"Rarely nucleogenic neon mimic cosmogenic neon isotopically⁵¹. In this case it can be completely excluded as the low U and Th concentrations in these quartz samples (<6 ppb; Dunai, et al. ¹⁵ supplementary data) produces several orders of magnitude less Ne than measured in these samples."

(ii) enhanced production rates due to higher paleoelevations, and (iii) prolonged transport times or long-distance sediment derivation from easterly source regions such as the uplifting Andes. These scenarios are systematically assessed and excluded using independent constraints and previously published data (e.g., Dunai et al., 2005). In particular, ^{21}Ne concentrations reported

from potential Andean source sediments (Azapa deposits) are significantly lower than those observed in our samples and therefore cannot explain the extreme values measured here.

See manuscript Line 179-199:

“Previous studies have suggested that inherited cosmogenic Ne might explain high cosmogenic nuclide concentrations^{19,52}. The only reasonable explanation by which the vein quartz fragments could have acquired high concentrations of cosmogenic ²¹Ne, other than by long exposure in an achingly slow evolving Coastal Cordillera landscape, is by shorter exposure at higher elevation. High altitude Andes-derived sediments could only have been delivered to the region prior to the Late Oligocene-Early Miocene when the Coastal Cordillera became isolated^{15,33-35}. Prior to isolation, the Western Cordillera did not exceed ~2.5 km elevation^{53,54} (Fig. 2C). If the old clasts were derived from the Western Cordillera, the cosmogenic ²¹Ne in excess of what was acquired after deposition at 23 Ma, would require exposure up to ~7 million years at the surface (Supplementary Datafile_1_TCN_Data). Such long exposure durations require extremely low erosion rates (<0.1 m/Myr, Supplementary Datafile_1_TCN_Data), which in turn implies extremely low rates of landscape change prior to the Miocene-Oligocene boundary in the uplifting Andes (and would be indicative of pre Miocene landscape stasis). However, such low erosion rates for the emerging Western Cordillera are highly unlikely^{25,53}. This is confirmed by low cosmogenic ²¹Ne concentrations measured by Dunai, et al. ¹⁵ in rounded quartz clasts from the Azapa Formation fluvial gravels that underlie the Jazpampa surfaces (Fig. 1, Site D in¹⁵, Extended Data Fig. 2). Two samples of amalgamated clasts (25 clasts each) yield cosmogenic ²¹Ne concentrations equivalent to less than 140 kyr of exposure (Supplementary Datafile_1_TCN_Data). The absence of pre-Miocene exposure ages in clasts from Precordillera alluvial fans¹⁷ is a further argument that the Andes have never shed long exposed sediment.”

In contrast, our additional sampling within the Coastal Cordillera reveals similarly high ²¹Ne concentrations, supporting a local signal consistent with in-situ cosmogenic production via spallation. Taken together, the combined dataset and analyses demonstrate that the measured ²¹Ne concentrations are best explained by prolonged surface exposure and cosmogenic production rather than by alternative processes.

Given these considerations, we are confident that the manuscript provides a thorough and critical evaluation of viable alternative interpretations. We would welcome more specific suggestions from the reviewer regarding additional scenarios that may warrant consideration.

Comment 2.2: In addition, no figure or map has been provided to visually synthesize the paleoclimatic forces and geological processes proposed to have contributed to aridity during the Miocene–Oligocene.

Response 2.2: In response to this comment, we have expanded Figure 3 to provide a visual synthesis of the key paleoclimatic and geological processes associated with the development of aridity. Specifically, we have added a new panel at the bottom of Figure 3 summarizing the major potential forcing factors contributing to the aridification of the Atacama Desert, as discussed in the manuscript.

We modified the figure caption and the link in the manuscript:

“Figure 3: Compilation of ^{21}Ne exposure ages and relevant regional and global paleo-datasets. (A) Relative probability plot of all new and previously published modelled cosmogenic Ne ages of clasts from the Jazpampa surfaces. Bars indicate qualitative reconstruction of surface activity and aridity state based on our data. (B) Compilation of supergene mineralization $^{39}\text{Ar}/^{40}\text{Ar}$ ages from the currently arid Precordillera and Western Cordillera of the Atacama Desert from Reich and Bao ⁷⁷ and proposed onsets of hyperaridity, shown by the black left-pointing arrows with citations at the top of the figure panel. (C) Andean uplift reconstructions after Garzione, et al. ⁵² for the Western and Eastern Cordillera and for the Andean Plateau (taken from Fig. 4 (central CAP 16-19°S) at 200, 400 and 500 km profile lengths) and uplift reconstructions for the Western Cordillera (20°S) from Scott, et al. ⁵³ and from Jimenez-Rodriguez, et al. ⁷⁸. (D) Oceanic gateway changes during the Neogene according to data compiled by Straume, et al. ⁶. Global Cenozoic reference benthic foraminifer oxygen

isotope dataset (CENOGRID) from Westerhold, et al. ⁵ covering the last 65 million years. The vertical grey bars show a qualitative representation of the ice volume in each hemisphere according to Westerhold, et al. ⁵. Major global climate periods/events are marked, climate states are defined by Westerhold, et al. ⁵. (E) Overview of the key potential drivers influencing the increasing aridity in the hyperarid core of the Atacama Desert. Global climate cooling since the EECO based on the Global Cenozoic reference benthic foraminifer oxygen isotope dataset (CENOGRID) from Westerhold, et al. ⁵. Increasing impact of the cold Humboldt Current with the gradual opening and deepening of the Drake Passage based on oceanic gateway changes during the Neogene according to data compiled by Straume, et al. ⁶ and increasing rain-shadow effect of the uplifting Andes based on data compilation from (B), with full impact of the recent rain-shadow effect by the uplifting Altiplano-Puna Plateau since the Mid Miocene.”

For these reasons, two essential revisions remain necessary before the manuscript can be considered for publication:

Comment 2.3:

1. Clarification of the study’s hypothetical nature.

In their rebuttal, the authors reiterate the arguments of the original manuscript while acknowledging the hypothetical character of the proposed model. Although the reasoning presented is plausible, it cannot be regarded as definitive. It is therefore essential that the tentative nature of the interpretation be explicitly conveyed in the title and abstract, rather than being confined to the discussion.

Response 2.3: In response to the reviewers comment, we made following changes to the abstract, Line 26-29:

“The long-term preservation of these rocks at the surface suggests an exceptionally low rate of landscape evolution and requires that extreme aridification of the region initiated considerably earlier than the development of the strong Humboldt Current and uplift of the Andes. We hypothesize that global cooling after the Early Eocene Climatic Optimum...”

We hope these revisions improve clarity and ensure that the interpretative nature of our model is evident from the outset. We also changed the title to meet the reviewers comment.

“Evidence for Eocene aridification of the Atacama Desert’s hyperarid core”

Comment 2.4:

2. Clear framing as a reassessment of previous work.

Although the authors demonstrate that their dataset substantially expands upon that of Dunai et al. (2005), the manuscript would benefit from a clearer statement in the introduction that the study constitutes a revisiting of the site. This contextualization would significantly aid readers.

Response 2.4: In response to the reviewers comment, we additionally now state at the end of the introduction paragraph within the brief summary of our work, that this works builds up on Dunai et al. 2005 and that we revisited the site again, plus additional sites in the area within this project. We modified Line 69-73:

“We also revisit the study site originally investigated by Dunai, et al. ¹⁵, substantially expanding their dataset by integrating previously published data with new ²¹Ne and ¹⁰Be measurements from multiple locations, as well as U-Pb zircon tephrochronology; in total, we compile concentrations from 135 ²¹Ne and seven ¹⁰Be surface clasts.”